# Explanation Shift:
# How Did the Distribution Shift Impact the Model?

**Carlos Mougan** *AI Office - European Commission & University of Southampton.*

**Klaus Broelemann** *Schufa Holding AG, Germany*

**Gjergji Kasneci** *Schufa Holding AG & Technical University of Munich*

**Thanassis Tiropanis** *University of Southampton*

**Steffen Staab** *University of Stuttgart & University of Southampton*

**Reviewed on OpenReview:** *https://openreview.net/forum?id=MO1slfU9xy*

## Abstract

The performance of machine learning models on new data is critical for their success in real-world applications. Current methods to detect shifts in the input or output data distributions have limitations in identifying model behavior changes when no labeled data is available. In this paper, we define *explanation shift* as the statistical comparison between how predictions from training data are explained and how predictions on new data are explained. We propose explanation shift as a key indicator to investigate the interaction between distribution shifts and learned models. We introduce an Explanation Shift Detector that operates on the explanation distributions, providing more sensitive and explainable changes in interactions between distribution shifts and learned models. We compare explanation shifts with other methods that are based on distribution shifts, showing that monitoring for explanation shifts results in more sensitive indicators for varying model behavior. We provide theoretical and experimental evidence and demonstrate the effectiveness of our approach on synthetic and real data. Additionally, we release an open-source Python package, `skshift`, which implements our method and provides usage tutorials for further reproducibility.

## 1 Introduction

Machine Learning (ML) theory provides means to forecast the quality of ML models on unseen data, provided that this data is sampled from the same distribution as the data used to train and evaluate the model. If unseen data is sampled from a different distribution than the training data, model quality may deteriorate, making monitoring how the model's behaviour changes crucial.

Recent research has highlighted the impossibility of reliably estimating the performance of machine learning models on unseen data sampled from a different distribution in the absence of further assumptions about the nature of the shift (Ben-David et al., 2010; Lipton et al., 2018; Garg et al., 2021). State-of-the-art techniques attempt to model statistical distances between the distributions of the training and unseen data or the distributions of the model predictions. However, these approaches only partially capture how *distribution shifts* affect the interaction between new data and trained models. Additionally, they often require causal graphs or specific shift assumptions, limiting their general applicability, mainly since such assumptions can depend on the data modality. In particular, tabular data—an economically critical domain—presents unique challenges. Thus, it is often necessary to go beyond detecting changes in input data distributions and understanding how they impact and relate to changes in the model given that performance degradation can not be accurately estimated.

The field of explainable AI has emerged as a way to understand model decisions and interpret the inner workings of ML models. The core idea of this paper is to go beyond the modeling of distribution shifts and monitor for *explanation shifts* to signal a change of interactions between learned models and dataset features in tabular data. We newly define explanation shift as the statistical comparison between how predictions from training data are explained and how predictions on new data are explained. In summary, our contributions are:

- We propose measures of explanation shifts as a key indicator for investigating the interaction between distribution shifts and learned models.

- We define an *Explanation Shift Detector* that operates on distributions of model explanations, allowing for more sensitive and explainable changes of interactions between distribution shifts and learned models in Section 3.

- We compare our monitoring method that is based on explanation shifts with methods that are based on other kinds of distribution shifts. We find that monitoring for explanation shifts results in more sensitive indicators for varying model behavior.

- We release an open-source Python package `skshift`, which implements our "*Explanation Shift Detector*", along usage tutorials for reproducibility (cf. Statement 6).

## 2 Foundations and Related Work

### 2.1 Basic Notions

Supervised machine learning induces a function $f_\theta : \text{dom}(X) \to \text{dom}(Y)$, from training data $\mathcal{D}^{tr} = \{(x_0^{tr}, y_0^{tr}), \ldots, (x_n^{tr}, y_n^{tr})\}$. Where dom(X) refers to the domain of $X$, which is the set of all possible values that the variable $X$ can take, and dom(Y) denotes the domain target of the target variable $Y$. Thereby, $f_\theta$ is from a family of functions $f_\theta \in F$ and $\mathcal{D}^{tr}$ is sampled from the joint distribution $\mathbf{P}(X, Y)$ with predictor variables $X$ and target variable $Y$. $f_\theta$ is expected to generalize well on new, previously unseen data $\mathcal{D}_X^{new} = \{x_0^{new}, \ldots, x_k^{new}\} \subseteq \text{dom}(X)$. We write $\mathcal{D}_X^{tr}$ to refer to $\{x_0^{tr}, \ldots, x_n^{tr}\}$ and $\mathcal{D}_Y^{tr}$ to refer to $\mathcal{D}_Y^{tr} = \{y_0^{tr} \ldots, y_n^{tr}\}$. For formalizations and to define evaluation metrics, it is often convenient to assume that an oracle provides values $\mathcal{D}_Y^{new} = \{y_0^{new}, \ldots, y_k^{new}\}$ such that $\mathcal{D}^{new} = \{(x_0^{new}, y_0^{new}), \ldots, (x_k^{new}, y_k^{new})\} \subseteq \text{dom}(X) \times \text{dom}(Y)$.

The core machine learning assumption is that training data $\mathcal{D}^{tr}$ and novel data $\mathcal{D}^{new}$ are sampled from the same underlying distribution $\mathbf{P}(X, Y)$. Where $\mathbf{P}(\cdot)$ are density functions for continuous variables or probability mass functions if the variables are discrete. The twin problems of *model monitoring* and recognizing that new data is *out-of-distribution* can now be described as predicting an absolute or relative performance drop between $\text{perf}(\mathcal{D}^{tr})$ and $\text{perf}(\mathcal{D}^{new})$, where $\text{perf}(\mathcal{D}) = \sum_{(x,y)\in\mathcal{D}} \ell_{\text{eval}}(f_\theta(x), y)$, $\ell_{\text{eval}}$ is a metric like 0-1-loss (accuracy), but $\mathcal{D}_Y^{new}$ is unknown and cannot be used for such judgment in an operating system.

Therefore, related work analyses distribution shifts between training and newly occurring data. Let two datasets $\mathcal{D}, \mathcal{D}'$ define two empirical distributions $\mathbf{P}(\mathcal{D}), \mathbf{P}(\mathcal{D}')$, then we write $\mathbf{P}(\mathcal{D}) \not\sim \mathbf{P}(\mathcal{D}')$ to express that $\mathbf{P}(\mathcal{D})$ is sampled from a different underlying distribution than $\mathbf{P}(\mathcal{D}')$.

**Definition 2.1** (Input Data Shift). We say that data shift occurs from $\mathcal{D}_X^{tr}$ to $\mathcal{D}_X^{new}$, if $\mathbf{P}(\mathcal{D}_X^{tr}) \not\sim \mathbf{P}(\mathcal{D}_X^{new})$.

Specific kinds of data shift are:

**Definition 2.2** (Univariate data shift). There is a univariate data shift between $\mathbf{P}(\mathcal{D}_X^{tr}) = \mathbf{P}(\mathcal{D}_{X_1}^{tr}, \ldots, \mathcal{D}_{X_p}^{tr})$ and $\mathbf{P}(\mathcal{D}_X^{new}) = \mathbf{P}(\mathcal{D}_{X_1}^{new}, \ldots, \mathcal{D}_{X_p}^{new})$, if $\exists i \in \{1 \ldots p\} : \mathbf{P}(\mathcal{D}_{X_i}^{tr}) \not\sim \mathbf{P}(\mathcal{D}_{X_i}^{new})$.

**Definition 2.3** (Covariate data shift). There is a covariate data shift between $\mathbf{P}(\mathcal{D}_X^{tr}) = \mathbf{P}(\mathcal{D}_{X_1}^{tr}, \ldots, \mathcal{D}_{X_p}^{tr})$ and $\mathbf{P}(\mathcal{D}_X^{new}) = \mathbf{P}(\mathcal{D}_{X_1}^{new}, \ldots, \mathcal{D}_{X_p}^{new})$ if $\mathbf{P}(\mathcal{D}_X^{tr}) \not\sim \mathbf{P}(\mathcal{D}_X^{new})$, which cannot only be caused by univariate shift.

The next two types of shift involve the interaction of data with the model $f_\theta$, which approximates the conditional $\frac{P(\mathcal{D}_Y^{tr}, \mathcal{D}_X^{tr})}{P(\mathcal{D}_X^{tr})}$. Abusing notation, we write $f_\theta(\mathcal{D})$ to refer to the multiset $\{f_\theta(x)|x \in \mathcal{D}\}$.

**Definition 2.4** (Predictions Shift)**.** There is a predictions shift between distributions $\mathbf{P}(\mathcal{D}_X^{tr})$ and $\mathbf{P}(\mathcal{D}_X^{new})$ related to model $f_\theta$ if $\mathbf{P}(f_\theta(\mathcal{D}_X^{tr})) \nsim \mathbf{P}(f_\theta(\mathcal{D}_X^{new}))$.

**Definition 2.5** (Concept Shift)**.** There is a concept shift between $\mathbf{P}(\mathcal{D}^{tr}) = \mathbf{P}(\mathcal{D}_X^{tr}, \mathcal{D}_Y^{tr})$ and $\mathbf{P}(\mathcal{D}^{new}) = \mathbf{P}(\mathcal{D}_X^{new}, \mathcal{D}_Y^{new})$ if conditional distributions change, i.e. $\frac{\mathbf{P}(\mathcal{D}_Y^{tr}, \mathcal{D}_X^{tr})}{\mathbf{P}(\mathcal{D}_X^{tr})} \nsim \frac{\mathbf{P}(\mathcal{D}_Y^{new}, \mathcal{D}_X^{new})}{\mathbf{P}(\mathcal{D}_X^{new})}$.

In practice, multiple types of shifts co-occur together, and their disentangling may constitute a significant challenge that we do not address here.

## 2.2 Related Work on Tabular Data

**Classifier two-sample test (C2ST):** Evaluating how two distributions differ has been a widely studied topic in the statistics and statistical learning literature (Hastie et al., 2001; Quiñonero-Candela et al., 2009; Liu et al., 2020a) and has advanced in recent years (Lee et al., 2018; Zhang et al., 2013). Using supervised learning classifiers to measure statistical tests has been explored by Lopez-Paz & Oquab (2017) proposing a classifier-based approach that returns test statistics to interpret differences between two distributions. We adopt their power test analysis and interpretability approach but apply it to the explanation distributions instead of input data distributions. Another noteworthy recent contribution comes from Barrabés et al. (2023), who leverages a tree-based classifier as C2ST. Their approach, augmented with iterative heuristics, aims at localizing and rectifying feature shifts within input data. In contrast, our work is distinctive in investigating the impact of distribution shifts on the model behaviour. We achieve this by applying C2ST to distributions of feature attribution explanations, providing insights into how alterations in distribution impact the model's predictive behavior.

Other methods to detect if new data is out-of-distribution (OOD) have relied on neural networks based on the prediction distributions Fort et al. (2021); Garg et al. (2020). They use the maximum softmax probabilities/likelihood as a confidence score Hendrycks & Gimpel (2017), temperature or energy-based scores Ren et al. (2019); Liu et al. (2020b); Wang et al. (2021), they extract information from the gradient space Huang et al. (2021), relying on the latent space Crabbé et al. (2021), they fit a Gaussian distribution to the embedding, or they use the Mahalanobis distance for out-of-distribution detection Lee et al. (2018); Park et al. (2021).

Many of these methods are explicitly developed for neural networks that operate on image and text data, and often, they can not be directly applied to traditional ML techniques. For example, in deep neural networks, one can define invariances in the latent space, which do not apply to tabular data. In this work, we focus on tabular data where techniques such as gradient-boosted decision trees achieve state-of-the-art model performance.

**Detecting distribution shift and its impact on model behaviour:** Extensive related work has aimed at detecting that data is from out-of-distribution. To this end, they have created several benchmarks that measure whether data comes from in-distribution or not (Malinin et al., 2021; Barrabés et al., 2023). In contrast, our main aim is to evaluate the impact of the distribution shift on the use of model. A typical example is two-sample testing on the latent space, as described by Rabanser et al. (2019). However, many methods developed for detecting out-of-distribution data are specific to neural networks processing image and text data and can not be applied to traditional machine learning techniques. These methods often assume that the relationships between predictor and response variables remain unchanged, i.e., no concept shift occurs. Our work is applied to tabular data where techniques such as gradient-boosted decision trees achieve state-of-the-art model performance (Grinsztajn et al., 2022; Elsayed et al., 2021; Borisov et al., 2021).

**Impossibility of model monitoring:** Recent research findings have formalized the limitations of monitoring machine learning models in the absence of labelled data. Specifically (Garg et al., 2021; Chen et al., 2022) prove the impossibility of predicting model degradation or detecting out-of-distribution data with certainty (Fang et al., 2022; Zhang et al., 2021; Guerin et al., 2022). Although our approach does not overcome these limitations, it provides valuable insights for machine learning engineers to better understand changes in interactions between learned models and shifting data distributions.

**Model monitoring and distribution shift under specific assumptions:** Under specific types of assumptions, model monitoring and distribution shift become feasible tasks. One type of assumption often found in the literature is to leverage causal knowledge to identify the drivers of distribution changes (Budhathoki et al., 2021; Zhang et al., 2022; Schrouff et al., 2022). For example, Budhathoki et al. (2021) use graphical causal models and feature attributions based on Shapley values to detect changes in the distribution. Similarly, other works aim to detect specific distribution shifts, such as covariate or concept shifts. Our approach does not rely on additional information, such as a causal graph, labelled test data, or specific types of distribution shift. Still, by the nature of pure concept shifts, the model behaviour remains unaffected and new data need to come with labelled responses to be detected.

**Explainability and distribution shift:** Lundberg et al. (2020) applied Shapley values to identify possible bugs in the pipeline by visualizing univariate SHAP contributions. Following this line of work, Nigenda et al. (2022) compare the order of the feature importance using the Normalized Discounted Cumulative Gain (NDCG) between training and unseen data. We go beyond their work and formalize the multivariate explanation distributions on which we perform a two-sample classifier test to detect how distribution shift impacts interaction with the model. Furthermore, we provide a mathematical analysis of how the SHAP values contribute to detecting distribution shift. In Appendix A we provide a formal comparison against Nigenda et al. (2022). Recent work by Kulinski & Inouye (2023) introduced a framework for explaining distribution shifts using a transport map between a source and target distribution; our work does not only change on the distribution shift but on how these changes impact the model.

## 2.3 Explainable AI: Local Feature Attributions

Attribution by Shapley values explains machine learning models by determining the relevance of features used by the model (Lundberg et al., 2020; Lundberg & Lee, 2017). The Shapley value is a concept from coalition game theory that aims to allocate the surplus generated by the grand coalition in a game to each of its players (Shapley, 1953). The Shapley value $\mathcal{S}_j$ for the $j$'th player is defined via a value function $\mathrm{val}_{f,x} : 2^N \to \mathbb{R}$ of players in $T$:

$$\mathcal{S}_j(\mathrm{val}_{f,x}) = \sum_{T \subseteq N \setminus \{j\}} \frac{|T|!(p - |T| - 1)!}{p!} (\mathrm{val}_{f,x}(T \cup \{j\}) - \mathrm{val}_{f,x}(T)), \tag{1}$$

$$\text{where} \quad \mathrm{val}_{f,x}(T) = E_{X|X_T = x_T}[f(X)] - E_X[f(X)]. \tag{2}$$

In machine learning, $N = \{1, \ldots, p\}$ is the set of features occurring in the training data. Given that $x$ is the feature vector of the instance to be explained, and the term $\mathrm{val}_{f,x}(T)$ represents the prediction for the feature values in $T$ that are marginalized over features that are not included in $T$. The Shapley value framework satisfies several theoretical properties (Molnar, 2019; Shapley, 1953; Winter, 2002; Aumann & Dreze, 1974). Our approach is based on the efficiency and uninformative properties:

**Efficiency Property.** Feature contributions add up to the difference of prediction from $x^\star$ and the expected value, $\sum_{j \in N} \mathcal{S}_j(f, x^\star) = f(x^\star) - E[f(X)])$.

**Uninformativeness Property.** A feature $j$ that does not change the predicted value has a Shapley value of zero. $\forall x, x_j, x'_j : f(\{x_{N \setminus \{j\}}, x_j\}) = f(\{x_{N \setminus \{j\}}, x'_j\}) \Rightarrow \forall x : \mathcal{S}_j(f, x) = 0$.

Our approach works with explanation techniques that fulfill efficiency and non-informative properties, and we use Shapley values as an example. It is essential to distinguish between the theoretical Shapley values and the different implementations that approximate them; in Appendix D, we provide an experimental comparison of different approaches.

LIME is another explanation method candidate for our approach (Ribeiro et al., 2016b;a) that can potentially satisfy efficiency and uninformative properties, even though several research has highlighted instability and difficulties with the definition of neighbourhoods. In Appendix C, we analyze LIME's relationship with Shapley values to describe explanation shifts.

## 3 A Model for Explanation Shift Detection

Our model for explanation shift detection is sketched in Fig. 1. We define it as follows:

**Definition 3.1** (Explanation distribution). An explanation function $\mathcal{S} : F \times \mathrm{dom}(X) \to \mathbb{R}^p$ maps a model $f_\theta \in F$ and data $x \in \mathbb{R}^p$ to a vector of attributions $\mathcal{S}(f_\theta, x) \in \mathbb{R}^p$. We call $\mathcal{S}(f_\theta, x)$ an explanation. We write $\mathcal{S}(f_\theta, \mathcal{D})$ to refer to the empirical *explanation distribution* generated by $\{\mathcal{S}(f_\theta, x) | x \in \mathcal{D}\}$.

We use local feature attribution methods SHAP and LIME as explanation functions $\mathcal{S}$.

**Definition 3.2** (Explanation shift). Given a model $f_\theta$ learned from $\mathcal{D}^{tr}$, explanation shift with respect to the model $f_\theta$ occurs if $\mathcal{S}(f_\theta, \mathcal{D}_X^{new}) \nsim \mathcal{S}(f_\theta, \mathcal{D}_X^{tr})$.

**Definition 3.3** (Explanation shift metrics). Given a measure of statistical distances $d$, explanation shift is measured as the distance between two explanations of the model $f_\theta$ by $d(\mathcal{S}(f_\theta, \mathcal{D}_X^{tr}), \mathcal{S}(f_\theta, \mathcal{D}_X^{new}))$.

We follow Lopez et al. (Lopez-Paz & Oquab, 2017) to define an explanation shift metrics based on a two-sample test classifier. We proceed as depicted in Figure 1. To counter overfitting, given the model $f_\theta$ trained on $\mathcal{D}^{tr}$, we compute explanations $\{\mathcal{S}(f_\theta, x) | x \in \mathcal{D}_X^{\mathrm{val}}\}$ on an in-distribution validation data set $\mathcal{D}_X^{\mathrm{val}}$. Given a dataset $\mathcal{D}_X^{new}$, for which the status of in- or out-of-distribution is unknown, we compute its explanations $\{\mathcal{S}(f_\theta, x) | x \in \mathcal{D}_X^{\mathrm{new}}\}$. Then, we construct a two-samples dataset $E = \{(S(f_\theta, x), a_x) | x \in \mathcal{D}_X^{\mathrm{val}}, a_x = 0\} \cup \{(S(f_\theta, x), a_x) | x \in \mathcal{D}_X^{\mathrm{new}}, a_x = 1\}$ and we train a discrimination model $g_\psi : \mathbb{R}^p \to \{0, 1\}$ on $E$, to predict if an explanation should be classified as in-distribution (ID) or out-of-distribution (OOD):

$$\psi = \arg\min_{\tilde{\psi}} \sum_{x \in \mathcal{D}_X^{\mathrm{val}} \cup \mathcal{D}_X^{\mathrm{new}}} \ell(g_{\tilde{\psi}}(\mathcal{S}(f_\theta, x)), a_x), \tag{3}$$

where $\ell$ is a classification loss function (e.g. cross-entropy). $g_\psi$ is our two-sample test classifier, based on which AUC yields a test statistic that measures the distance between the $D_X^{tr}$ explanations and the explanations of new data $D_X^{new}$.

Explanation shift detection allows us to detect *that* a novel dataset $D^{new}$ changes the model's behavior. Beyond recognizing explanation shift, using feature attributions for the model $g_\psi$, we can interpret *how* the features of the novel dataset $D_X^{new}$ interact differently with model $f_\theta$ than the features of the validation dataset $D_X^{val}$. These features are to be considered for model monitoring and for classifying new data as out-of-distribution.

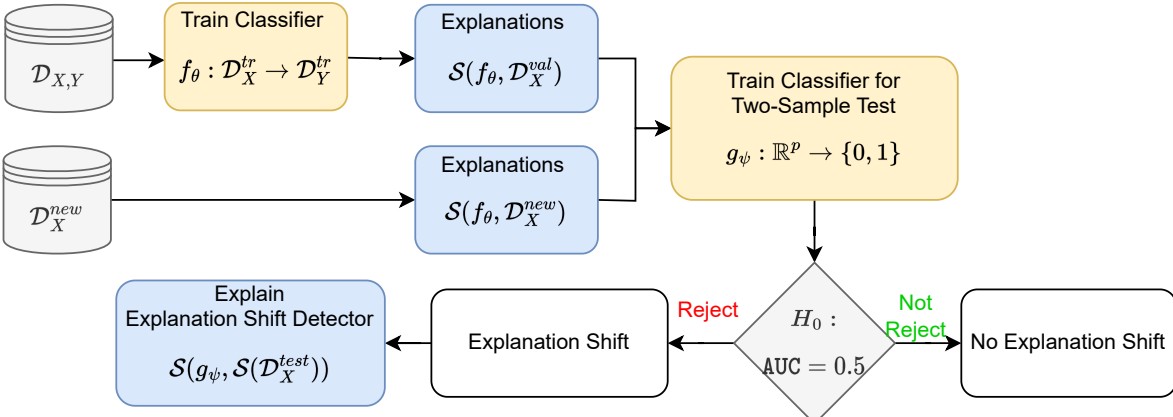

**Figure 1:** Our model for explanation shift detection. The model $f_\theta$ is trained on $\mathcal{D}^{tr}$ implying explanations for distributions $\mathcal{D}_X^{val}, \mathcal{D}_X^{new}$. The AUC of the two-sample test classifier $g_\psi$ decides for or against explanation shift. If an explanation shift occurred, it could be explained which features of the $\mathcal{D}_X^{new}$ deviated in $f_\theta$ compared to $\mathcal{D}_X^{val}$.

# 4 Relationships between Common Distribution Shifts and Explanation Shifts

This section analyses and compares data shifts and prediction shifts with explanation shifts. Section 4.4 draws from these analyses to derive experiments with synthetic data.

## 4.1 Explanation Shift vs Data Shift

One type of distribution shift that is challenging to detect comprises cases where the univariate distributions for each feature $j$ are equal between the source $\mathcal{D}_X^{tr}$ and the unseen dataset $\mathcal{D}_X^{new}$, but where interdependencies among different features change. Multi-covariance statistical testing is a hard task with high sensitivity that can lead to false positives. The following example demonstrates that Shapley values can indicate co-variate interaction changes while a univariate statistical test will provide false negatives.

**Example 4.1.** *(**Multivariate Shift**) Let $D^{tr} \sim N\left(\begin{bmatrix} \mu_1 \\ \mu_2 \end{bmatrix}, \begin{bmatrix} \sigma_{X_1}^2 & 0 \\ 0 & \sigma_{X_2}^2 \end{bmatrix}\right) \times Y$. We fit a linear model* $f_\theta(x_1, x_2) = \gamma + a \cdot x_1 + b \cdot x_2$. *If* $\mathcal{D}_X^{new} \sim N\left(\begin{bmatrix} \mu_1 \\ \mu_2 \end{bmatrix}, \begin{bmatrix} \sigma_{X_1}^2 & \rho\sigma_{X_1}\sigma_{X_2} \\ \rho\sigma_{X_1}\sigma_{X_2} & \sigma_{X_2}^2 \end{bmatrix}\right)$, *then* $\mathbf{P}(\mathcal{D}_{X_1}^{tr})$ *and* $\mathbf{P}(\mathcal{D}_{X_2}^{tr})$ *are identically distributed with* $\mathbf{P}(\mathcal{D}_{X_1}^{new})$ *and* $\mathbf{P}(\mathcal{D}_{X_2}^{new})$, *respectively, while this does not hold for the corresponding* $\mathcal{S}_j(f_\theta, \mathcal{D}_X^{tr})$ *and* $\mathcal{S}_j(f_\theta, \mathcal{D}_X^{new})$ *for* $j \in \{1, 2\}$.

$$\mathcal{S}_1(f_\theta, x) = a \cdot (x_1 - \mu_1),$$
$$\mathcal{S}_1(f_\theta, x^{new}) = \frac{1}{2}\left[\text{val}(\{1, 2\}) - \text{val}(\{2\})\right] + \frac{1}{2}\left[\text{val}(\{1\}) - \text{val}(\emptyset)\right],$$

where the Shapley value function val is defined as follows:

$$\text{val}(\{1, 2\}) = \mathbb{E}[f_\theta | X_1 = x_1, X_2 = x_2] = a \cdot x_1 + b \cdot x_2,$$
$$\text{val}(\emptyset) = \mathbb{E}[f_\theta] = a \cdot \mu_1 + b \cdot \mu_2,$$
$$\text{val}(\{1\}) = \mathbb{E}[f_\theta | X_1 = x_1] + b \cdot \mu_2,$$
$$= a \cdot \mu_1 + \rho \cdot \frac{\sigma_{X_2}}{\sigma_{X_1}} \cdot (x_1 - \mu_1) + b \cdot \mu_2,$$
$$\text{val}(\{2\}) = \mathbb{E}[f_\theta | X_2 = x_2] + a \cdot \mu_1,$$
$$= b \cdot \mu_2 + \rho \cdot \frac{\sigma_{X_1}}{\sigma_{X_2}} \cdot (x_2 - \mu_2) + a \cdot \mu_1.$$

Substituting these values, we conclude:

$$\mathcal{S}_1(f_\theta, x^{new}) \neq a \cdot (x_1 - \mu_1).$$

False positives frequently occur in out-of-distribution data detection when a statistical test recognizes differences between a source distribution and a new distribution, though the differences do not affect the model behavior (Grinsztajn et al., 2022; Huyen, 2022). Shapley values satisfy the *Uninformativeness* property, where a feature $j$ that does not change the predicted value has a Shapley value of 0 (equation 2.3).

**Example 4.2.** *Shifts on Uninformative Features. Let the random variables $X_1, X_2$ be normally distributed with $N(0; 1)$. Let dataset $\mathcal{D}^{tr} \sim X_1 \times X_2 \times Y^{tr}$, with $Y^{tr} = X_1$. Thus $Y^{tr} \perp X_2$. Let $\mathcal{D}_X^{new} \sim X_1 \times X_2^{new}$ and $X_2^{new}$ be normally distributed with $N(\mu; \sigma^2)$ and $\mu, \sigma \in \mathbb{R}$. When $f_\theta$ is trained optimally on $\mathcal{D}^{tr}$ then $f_\theta(x) = x_1$. $\mathbf{P}(\mathcal{D}_{X_2}^{tr})$ is different from $\mathbf{P}(\mathcal{D}_{X_2}^{new})$ but $\mathcal{S}_2(f_\theta, \mathcal{D}_X^{tr}) = 0 = \mathcal{S}_2(f_\theta, \mathcal{D}_X^{new})$.*

$$\mathcal{D}_{X_3} \sim N(\mu_3, c_3), \mathcal{D}_{X_3}^{new} \sim N(\mu_3^{'}, c_3^{'}).$$
$$\text{If} \quad \mu_3^{'} \neq \mu_3 \quad \text{or} \quad c_3^{'} \neq c_3 \rightarrow P(X_3) \neq P(X_3^{new}),$$
$$\mathcal{S}(f_\theta, X) = \left( \begin{bmatrix} a_1(X_1 - \mu_1) \\ a_2(X_2 - \mu_2) \\ a_3(X_3 - \mu_3) \end{bmatrix} \right) = \left( \begin{bmatrix} a_1(X_1 - \mu_1) \\ a_2(X_2 - \mu_2) \\ 0 \end{bmatrix} \right).$$
$$\mathcal{S}_3(f_\theta, \mathcal{D}_X) = \mathcal{S}_3(f_\theta, \mathcal{D}_X^{new}).$$

### 4.2 Explanation Shift vs Prediction Shift

Analyses of the explanations detect distribution shifts that interact with the model. In particular, if a prediction shift occurs, the explanations produced are also shifted.

**Proposition 1.** Given a model $f_\theta : \mathcal{D}_X \rightarrow \mathcal{D}_Y$. If $f_\theta(x') \neq f_\theta(x)$, then $\mathcal{S}(f_\theta, x') \neq \mathcal{S}(f_\theta, x)$.

This follows from the efficiency property of Shapley values (Aas et al., 2021). For example, given:

$$\texttt{Given} \quad f_\theta(x) \neq f_\theta(x') \sum_{j=1}^{p} \mathcal{S}_j(f_\theta, x) = f_\theta(x) - E_X[f_\theta(\mathcal{D}_X)] \texttt{ then } \quad \mathcal{S}(f_\theta, x) \neq \mathcal{S}(f_\theta, x'), \tag{4}$$

a difference in predictions implies at least one component of the explanation vectors must differ.

However, the reverse is not necessarily true: an explanation shift does not always imply a prediction shift. The following example illustrates this asymmetry.

**Example 4.3.** *(**Explanation shift not affecting prediction distribution**) Given $\mathcal{D}^{tr}$ is generated from $(X_1 \times X_2 \times Y), X_1 \sim U(0,1), X_2 \sim U(1,2), Y = X_1 + X_2 + \epsilon$ and thus the optimal model is $f(x) = x_1 + x_2$. If $\mathcal{D}^{new}$ is generated from $X_1^{new} \sim U(1,2), X_2^{new} \sim U(0,1), \quad Y^{new} = X_1^{new} + X_2^{new} + \epsilon$, the prediction distributions are identical $f_\theta(\mathcal{D}_X^{tr}), f_\theta(\mathcal{D}_X^{new}) \sim U(1,3)$, but explanation distributions are different $S(f_\theta, \mathcal{D}_X^{tr}) \nsim S(f_\theta, \mathcal{D}_X^{new})$, because $\mathcal{S}_i(f_\theta, x) = \alpha_i \cdot x_i$. for $i \in \{1,2\}$.*

$$\forall i \in \{1,2\} \quad \mathcal{S}_i(f_\theta, x) = \alpha_i \cdot x_i,$$
$$\mathcal{S}_i(f_\theta, \mathcal{D}_X)) \neq \mathcal{S}_i(f_\theta, \mathcal{D}_X^{new}),$$
$$\Rightarrow f_\theta(\mathcal{D}_X) = f_\theta(\mathcal{D}_X^{new}).$$

### 4.3 Explanation Shift vs Concept Shift

Concept shift comprises cases where the covariates retain a given distribution, but their relationship with the target variable changes (cf. Section 2.1). This example shows the negative result that the detection of explanation shift cannot indicate concept shift.

**Example 4.4.** ***Concept Shift*** *Let $\mathcal{D}^{tr} \sim X_1 \times X_2 \times Y$, and create a synthetic target $y_i^{tr} = a_0 + a_1 \cdot x_{i,1} + a_2 \cdot x_{i,2} + \epsilon$. As new data we have $\mathcal{D}_X^{new} \sim X_1^{new} \times X_2^{new} \times Y$, with $y_i^{new} = b_0 + b_1 \cdot x_{i,1} + b_2 \cdot x_{i,2} + \epsilon$ whose coefficients are unknown at prediction stage. With coefficients $a_0 \neq b_0, a_1 \neq b_1, a_2 \neq b_2$. We train a linear regression $f_\theta : \mathcal{D}_X^{tr} \rightarrow \mathcal{D}_Y^{tr}$. Then explanations have the same distribution, $\mathbf{P}(\mathcal{S}(f_\theta, \mathcal{D}_X^{tr})) = \mathbf{P}(\mathcal{S}(f_\theta, \mathcal{D}_X^{new}))$, input data distribution $\mathbf{P}(\mathcal{D}_X^{tr}) = \mathbf{P}(\mathcal{D}_X^{new})$ and predictions $\mathbf{P}(f_\theta(\mathcal{D}_X^{tr})) = \mathbf{P}(f_\theta(\mathcal{D}_X^{new}))$. But there is no guarantee on the performance of $f_\theta$ on $\mathcal{D}_X^{new}$ (Garg et al., 2021).*

$$X \sim N(\mu, \sigma^2 \cdot I), X^{new} \sim N(\mu, \sigma^2 \cdot I),$$
$$\rightarrow P(\mathcal{D}_X) = P(\mathcal{D}_X^{new}),$$
$$Y \sim a + \alpha N(\mu, \sigma^2) + \beta N(\mu, \sigma^2) + N(0, \sigma'^2),$$
$$Y^{new} \sim a + \beta N(\mu, \sigma^2) + \alpha N(\mu, \sigma^2) + N(0, \sigma'^2),$$
$$\rightarrow P(\mathcal{D}_Y) = P(\mathcal{D}_Y^{new}).$$
$$\mathcal{S}(f_\theta, \mathcal{D}_X) = \begin{pmatrix} \alpha(X_1 - \mu_1) \\ \beta(X_2 - \mu_2) \end{pmatrix} \sim \begin{pmatrix} N(\mu_1, \alpha^2 \sigma^2) \\ N(\mu_2, \beta^2 \sigma^2) \end{pmatrix}.$$
$$\mathcal{S}(h_\phi, \mathcal{D}_X) = \begin{pmatrix} \beta(X_1 - \mu_1) \\ \alpha(X_2 - \mu_2) \end{pmatrix} \sim \begin{pmatrix} N(\mu_1, \beta^2 \sigma^2) \\ N(\mu_2, \alpha^2 \sigma^2) \end{pmatrix}.$$
$$\text{If} \quad \alpha \neq \beta \rightarrow \mathcal{S}(f_\theta, \mathcal{D}_X) \neq \mathcal{S}(h_\phi, \mathcal{D}_X).$$

In general, concept shift cannot be detected because $\mathcal{D}_Y^{new}$ is unknown (Garg et al., 2021). Some research studies have made specific assumptions about the conditional $\frac{P(\mathcal{D}_Y^{new}), \mathcal{D}_X^{new}}{P(\mathcal{D}_X^{new})}$ in order to monitor models and detect distribution shift (Lu et al., 2023; Alvarez et al., 2023). Appendix 4.3 sketches a situation where explanation distributions are used in the context of labelled data to indicate concept shift— which is not the paper's main target.

### 4.4 Experiments on Synthetic Data

This experimental section explores the detection of distribution shifts via explanation shifts in the previous analytical examples. In this case, the model is non-linear, a gradient-boosted decision tree from the `XGBoost` Python implementation.

#### 4.4.1 Detecting Multivariate Shift

Given two bivariate normal distributions $\mathcal{D}_X = (X_1, X_2) \sim N\left(0, \begin{bmatrix} 1 & 0 \\ 0 & 1 \end{bmatrix}\right)$ and $\mathcal{D}_X^{new} = (X_1^{new}, X_2^{new}) \sim N\left(0, \begin{bmatrix} 1 & 0.2 \\ 0.2 & 1 \end{bmatrix}\right)$, then, for each feature $j$ the underlying distribution is equally distributed between $\mathcal{D}_X$ and $\mathcal{D}_X^{new}$, $\forall j \in \{1, 2\} : P(\mathcal{D}_{X_j}) = P(\mathcal{D}_{X_j}^{new})$, and what is different are the interaction terms between them. We now create a synthetic target $Y = X_1 \cdot X_2 + \epsilon$ with $\epsilon \sim N(0, 0.1)$ and fit a gradient-boosted decision tree $f_\theta(\mathcal{D}_X)$. Then we compute the SHAP explanation values for $\mathcal{S}(f_\theta, \mathcal{D}_X)$ and $\mathcal{S}(f_\theta, \mathcal{D}_X^{new})$.

**Table 1:** Displayed results are the one-tailed p-values of the Kolmogorov-Smirnov (KS) test comparison between two underlying distributions. Small p-values indicate that compared distributions are unlikely to be equally distributed. SHAP values correctly indicate the interaction changes that individual distribution comparisons cannot detect.

| Comparison | p-value | Conclusions |
|---|---|---|
| $\mathbf{P}(\mathcal{D}_{X_1}), \mathbf{P}(\mathcal{D}_{X_1}^{new})$ | 0.33 | Not Distinct |
| $\mathbf{P}(\mathcal{D}_{X_2}), \mathbf{P}(\mathcal{D}_{X_2}^{new})$ | 0.60 | Not Distinct |
| $\mathcal{S}_1(f_\theta, \mathcal{D}_X), \mathcal{S}_1(f_\theta, \mathcal{D}_X^{new})$ | 3.9e−153 | Distinct |
| $\mathcal{S}_2(f_\theta, \mathcal{D}_X), \mathcal{S}_2(f_\theta, \mathcal{D}_X^{new})$ | 2.9e−148 | Distinct |

Having drawn $50,000$ samples from both $\mathcal{D}_X$ and $\mathcal{D}_X^{new}$, in Table 1, we evaluate whether changes in the input data distribution or the explanations can detect changes in covariate distribution. For this, we compare the one-tailed p-values of the Kolmogorov-Smirnov test between the input data distribution and the explanations distribution. The "Distinct/Not Distinct" conclusion is based on the one-tailed p-value of the Kolmogorov-Smirnov test drawn out of $50,000$ samples for both distributions, this comparison methodology is used similarly for the rest of the experiments on synthetic data section 5.2. Explanation shift correctly detects the multivariate distribution change that univariate statistical testing can not detect.

### 4.4.2 Detecting Concept Shift

As mentioned before, concept shifts cannot be detected if new data comes without target labels. However, if new data is labelled, the explanation shift can still be useful for detecting concept shifts.

Given a bivariate normal distribution $\mathcal{D}_X = (X_1, X_2) \sim N(1, I)$ where $I$ is an identity matrix of order two. We now create two synthetic targets $Y = X_1^2 \cdot X_2 + \epsilon$ and $Y^{new} = X_1 \cdot X_2^2 + \epsilon$ and fit two machine learning models $f_\theta : \mathcal{D}_X \to \mathcal{D}_Y$ and $h_\Upsilon : \mathcal{D}_X \to \mathcal{D}_Y^{new}$. Now we compute the SHAP values for $\mathcal{S}(f_\theta, \mathcal{D}_X)$ and $\mathcal{S}(h_\Upsilon, \mathcal{D}_X)$.

**Table 2:** Comparison of distribution shifts in synthetic concept shift analysis, highlighting the distinctiveness of SHAP value distributions in detecting relational changes between features and targets, even when other distributions appear equivalent.

| Comparison | Conclusions |
|---|---|
| $\mathbf{P}(\mathcal{D}_X), \mathbf{P}(\mathcal{D}_X^{new})$ | Not Distinct |
| $\mathbf{P}(\mathcal{D}_Y), \mathbf{P}(\mathcal{D}_Y^{new})$ | Not Distinct |
| $\mathbf{P}(f_\theta(\mathcal{D}_X)), \mathbf{P}(h_\Upsilon(\mathcal{D}_X^{new}))$ | Not Distinct |
| $\mathbf{P}(\mathcal{S}(f_\theta, \mathcal{D}_X)), \mathbf{P}(\mathcal{S}(h_\Upsilon, \mathcal{D}_X))$ | Distinct |

In Table 2, we see how the distribution shifts cannot capture the change in the model behavior while the SHAP values are different. As in the synthetic example, in table 2, SHAP values can detect a relational change between $\mathcal{D}_X$ and $\mathcal{D}_Y$, even if both distributions remain equivalent.

### 4.4.3 Uninformative Features on Synthetic Data

To have an applied use case of the synthetic example from the methodology section, we create a three-variate normal distribution $\mathcal{D}_X = (X_1, X_2, X_3) \sim N(0, I_3)$, where $I_3$ is an identity matrix of order three. The target variable is generated $Y = X_1 \cdot X_2 + \epsilon$ being independent of $X_3$. For both training and test data, $50,000$ samples are drawn. Then out-of-distribution data is created by shifting $X_3$, which is independent of the target, on test data $\mathcal{D}_{X_3}^{new} = \mathcal{D}_{X_3}^{te} + 1$.

**Table 3:** Distribution comparison when modifying a random noise variable on test data. The input data shifts while explanations and predictions do not.

| Comparison | Conclusions |
|---|---|
| $\mathbf{P}(\mathcal{D}_{X_3}^{te}), \mathbf{P}(\mathcal{D}_{X_3}^{new})$ | Distinct |
| $f_\theta(\mathcal{D}_X^{te}), f_\theta(\mathcal{D}_X^{new})$ | Not Distinct |
| $\mathcal{S}(f_\theta, \mathcal{D}_X^{te}), \mathcal{S}(f_\theta, \mathcal{D}_X^{new})$ | Not Distinct |

In Table 3, we see how an unused feature has changed the input distribution, but the explanation distributions and performance evaluation metrics remain the same.

### 4.4.4 Explanation Shift that does not affect the Prediction

In this case, we provide a situation where we have changes in the input data distributions that affect the model explanations but do not affect the model predictions because positive and negative associations between the model predictions and the distributions cancel out, producing a vanishing correlation in the mixture of the distribution (Yule effect 4.2).

We create a train and test data by drawing $50,000$ samples from a bi-uniform distribution $X_1 \sim U(0, 1), \quad X_2 \sim U(1, 2)$ the target variable is generated by $Y = X_1 + X_2$ where we train our model $f_\theta$. Then if out-of-distribution data is sampled from $X_1^{new} \sim U(1, 2), X_2^{new} \sim U(0, 1)$.

In Table 4, we see how an unused feature has changed the input distribution, but the explanation distributions and performance evaluation metrics remain the same.

**Table 4:** Distribution comparison over how the change on the contributions of each feature can cancel out to produce an equal prediction (cf. Section 4.2), while explanation shift will detect this behaviour changes on the predictions will not.

| Comparison | Conclusions |
|---|---|
| $f(\mathcal{D}_X^{te})$, $f(\mathcal{D}_X^{new})$ | Not Distinct |
| $\mathcal{S}(f_\theta, \mathcal{D}_{X_2}^{te})$, $\mathcal{S}(f_\theta, \mathcal{D}_{X_2}^{new})$ | Distinct |
| $\mathcal{S}(f_\theta, \mathcal{D}_{X_1}^{te})$, $\mathcal{S}(f_\theta, \mathcal{D}_{X_1}^{new})$ | Distinct |

## 4.5 Summary Comparison on Synthetic data

To assess the effectiveness of different detection methods in identifying and accounting for synthetic shifts, we present a conceptual comparison in Table 5. We evaluate these methods based on their capacity to capture synthetic shifts. We illustrate this comparison by considering two scenarios: a multicovariate shift (cf. Example 4.1) and a shift involving uninformative features (cf. Example 4.2). The complementary evaluation with related work is in the Appendix in Section A

This comparison focuses on their ability to detect synthetic distribution shifts using the examples of covariate shifts and uninformative shifts. It provides valuable insights while ensuring accountability.

**Table 5:** Conceptual comparison of different detection methods over the examples discussed in the mathematical analysis of the main body of the paper(cf. Section 4): a multicovariate shift(cf. Example 4.1 )and an uninformative features shift(cf. Example 4.2) . Learning a Classifier Two-Sample test $g$ over the explanation distributions is the only method that achieves the desired results (✓) and is accountable. We evaluate accountability by checking if the feature attributions of the detection method correspond to the synthetic shift generated in both scenarios

| Detection Method | Covariate | Uninformative | Accountability |
|---|:---:|:---:|:---:|
| Input distribution($g_\phi$) | ✓ | ✗ | ✗ |
| Prediction distribution($g_\Upsilon$) | ✓ | ✓ | ✗ |
| Input KS | ✗ | ✗ | ✗ |
| Classifier Drift | ✓ | ✗ | ✗ |
| Output KS | ✓ | ✓ | ✗ |
| Output Wasserstein | ✓ | ✓ | ✗ |
| Uncertainty | ~ | ✓ | ✓ |
| NDCG | ✗ | ✓ | ✗ |
| Explanation distribution ($g_\psi$) Explanation Shift Detector | ✓ | ✓ | ✓ |

# 5 Empirical Evaluation

We evaluate the effectiveness of explanation shift detection on tabular data by comparing it against methods from the literature, which are all based on discovering distribution shifts. For this comparison, we systematically vary models $f$, model parameterizations $\theta$, and input data distributions $\mathcal{D}_X$. The experimental results include:

1. Details on experiments with synthetic data (cf. Section 5.2).

2. Experiments on natural datasets (cf. Sections 5.3, 5.4, and further extended on Appendix B).

3. A broad range of modeling choices for both the model $f$ and the detector $g$ (cf. Section 5.5).

4. Analysis of the effects of hyperparameter variations on explanation shifts for the Explanation Shift Detector (Appendix B.5) and the estimator (cf. Section 5.6).

5. A comparison of our SHAP-based method against LIME, an alternative explanation approach (Appendix C).

Core observations made in this section are further confirmed and refined in the Appendix, without being contradicted.

### 5.1 Baseline Methods and Datasets

**Baseline Methods.** We compare our method of explanation shift detection (Section 3) with several methods that aim to detect that input data is out-of-distribution: *(B1)* statistical Kolmogorov Smirnov (KS) test on input data (Rabanser et al., 2019), *(B2)* prediction shift detection by Wasserstein distance (Lu et al., 2023), *(B3)* NDCG-based test of feature importance between the two distributions (Nigenda et al., 2022), *(B4)* prediction shift detection by Kolmogorov-Smirnov test (Diethe et al., 2019), and *(B5)* model agnostic uncertainty estimation (Mougan & Nielsen, 2023; Kim et al., 2020). All Distribution Shift Metrics are scaled between 0 and 1. We also compare against Classifier Two-Sample Test (Lopez-Paz & Oquab, 2017) on different distributions as discussed in Section 4, viz. *(B6)* classifier two-sample test on input distributions ($g_\phi$) following Barrabés et al. (2023) and *(B7)* classifier two-sample test on the predictions distributions ($g_\Upsilon$):

$$\phi = \arg\min_{\tilde{\phi}} \sum_{x \in \mathcal{D}_X^{val} \cup \mathcal{D}_X^{new}} \ell(g_{\tilde{\phi}}(x)), a_x). \tag{5}$$

$$\Upsilon = \arg\min_{\tilde{\Upsilon}} \sum_{x \in \mathcal{D}_X^{val} \cup \mathcal{D}_X^{new}} \ell(g_{\tilde{\Upsilon}}(f_\theta(x)), a_x). \tag{6}$$

**Datasets.** In the main body of the paper we base our comparisons on the UCI Adult Income dataset Dua & Graff (2017) and on synthetic data. In the Appendix, we extend experiments to several other datasets, which confirm our findings: ACS Travel Time, ACS Employment, Stackoverflow dataset (Stackoverflow, 2019).

### 5.2 Synthetic Data

Our first experiment on synthetic data showcases the two main contributions of our method: (*i*) being more sensitive to changes in the model than prediction shift and input shift and (*ii*) accounting for its drivers. We first generate a synthetic dataset $\mathcal{D}^\rho$, with a parametrized multivariate shift between $(X_1, X_2)$, where $\rho$ is the correlation coefficient, and an extra variable $X_3 = N(0, 1)$ and generate our target $Y = X_1 \cdot X_2 + X_3$. We train the $f_\theta$ on $\mathcal{D}^{tr, \rho=0}$ using a gradient-boosted decision tree, while for $g_\psi : \mathcal{S}(f_\theta, \mathcal{D}_X^{val, \rho}) \to \{0, 1\}$, we train on different datasests with different values of $\rho$. For $g_\psi$ we use a logistic regression. In Section 5.5, we benchmark other models $f_\theta$ and detectors $g_\psi$.

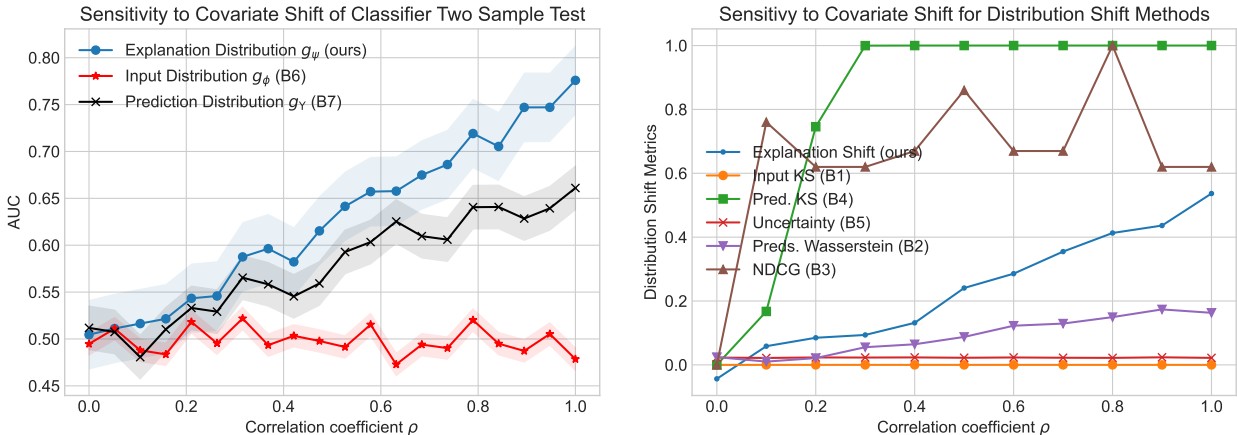

**Figure 2:** In the left figure, we compare the Classifier Two-Sample Test on explanation distribution (ours) versus input distribution *(B6)* and prediction distribution *(B7)*. Explanation distribution shows the highest sensitivity. The right figure, related work comparison of distribution shift methods*(B1-B5)*, as the experimental setup has a gradual distribution shift, good indicators should follow a progressive steady positive slope, following the correlation coefficient, as our method does. In Table 6 we provide a quantitative evaluation.

The left image in Figure 2 compares our approach against C2ST on input data distribution*(B6)* and on the predictions distribution *(B7)* different data distributions, for detecting multi-covariate shifts on different distributions. In our covariate experiment, we observed that using the explanation shift led to higher

sensitivity towards detecting distribution shift. We interpret the results with the efficiency property of the Shapley values, which decomposes the vector $f_\theta(\mathcal{D}_X)$ into the matrix $\mathcal{S}(f_\theta, \mathcal{D}_X)$. Moreover, we can identify the features that cause the drift by extracting the coefficients of $g_\psi$, providing global and local explainability.

The right image features the same setup compared to the other out-of-distribution detection methods (B1-B5). Table 6 quantitatively evaluates how the baselines correlate with the covariate correlation coefficient ($\rho$). One can see how our method behaves favourably compared to the others.

**Table 6:** Pearson Correlation of the correlation coefficient $\rho$ and baseline methods, extending Figure 2. Explanation Shift achieves better covariate shift detection on synthetic data.

| Baseline | Pearson Correlation with $\rho$ |
|---|---|
| B1 Input KS | 0.01 |
| B2 Prediction Wasserstein | 0.97 |
| B3 Explanation NDCG | 0.52 |
| B4 Prediction KS | 0.70 |
| B5 Uncertainty | 0.26 |
| B6 C2ST Input | 0.18 |
| B7 C2ST Output | 0.96 |
| (Ours) Explanation Shift | **0.99** |

### 5.3 Novel Group Shift

The distribution shift in this experimental setup is constituted by the appearance of a hitherto unseen group at prediction time (the group information is not present in the training features). We vary the ratio of presence of this unseen group in $\mathcal{D}_X^{new}$ data. The experiment is done with two $f_\theta$ models: a gradient-boosted decision tree and a logistic regression; for $g_\psi$, we use a logistic regression. Results are presented in Figure 3 and Table 7. Confidence intervals are extracted out of 10 bootstraps. Furthermore, we compare the performance of different algorithms for $f_\theta$ and $g_\psi$ in Section 5.5, and varying hyperparameters in Section 5.6.

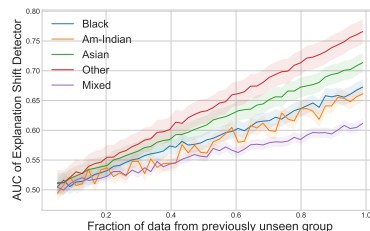 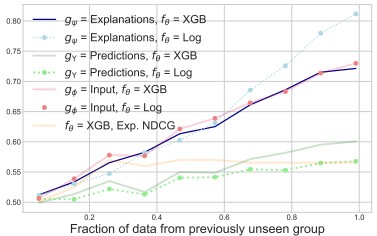 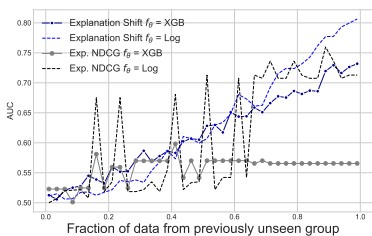

**Figure 3:** Novel group shift experiment on the US Income dataset. Sensitivity (AUC) increases with the growing fraction of previously unseen social groups. *Left figure*: The explanation shift indicates that different social groups exhibit varying deviations from the distribution on which the model was trained (White). *Middle Figure*: We vary the model $f_\theta$ by training it using both `XGBoost` (solid lines) and Logistic Regression (dots). The novel ethnicity group is Black. We compare Explanation Shift against C2ST on input *(B6)* and output *(B7)*. *Right figure*: Comparison of Explanation Shift against Exp. NDCG *(B4)*. We see how monitoring method *(B4)* is more unstable with a linear model, and with an `XGBoost` it erroneously finds a horizontal asymptote. We don't compare against methods relying purely on input data such as *(B1)* as we are changing the model, which they don't take into consideration.

### 5.4 Geopolitical and Temporal Shift

In this section, we tackle a geopolitical and temporal distribution shift; for this, the training data $\mathcal{D}^{tr}$ for the model $f_\theta$ is composed of data from California in 2014 and a $\mathcal{D}^{new}$ for each of the states in 2018. The objective of the prediction task of this dataset is to predict whether an individual's income is above \$50,000.[1].

---

[1]For more information consult `https://github.com/socialfoundations/folktables` of the associated publication Ding et al. (2021b)

**Table 7:** Pearson correlation between baselines and the ratio of presence of the unseen group. The *Accountable* column indicates accountability, measured as providing both theoretical underpinnings and empirical validation of the sources driving model changes, as discussed in Section 4.

| Baseline | Pearson Correlation | | Accountable |
|---|---|---|---|
| | $f_\theta = \textbf{Log}$ | $f_\theta = \textbf{XGB}$ | |
| B1 Input KS | $\mathbf{0.99 \pm 0.01}$ | $\mathbf{0.99 \pm 0.01}$ | ✗ |
| B2 Pred. Wass. | $0.95 \pm 0.02$ | $\mathbf{0.98 \pm 0.01}$ | ✗ |
| B3 NDCG | $0.37 \pm 0.25$ | $0.81 \pm 0.10$ | ✗ |
| B4 Pred. KS | $0.97 \pm 0.02$ | $0.96 \pm 0.01$ | ✗ |
| B5 Uncertainty | $0.73 \pm 0.10$ | $0.74 \pm 0.12$ | ✓ |
| B6 C2ST Input | $0.95 \pm 0.03$ | $0.95 \pm 0.03$ | ✗ |
| B7 C2ST Output | $0.67 \pm 0.13$ | $0.96 \pm 0.02$ | ✗ |
| Explanation Shift | $\mathbf{0.98 \pm 0.01}$ | $\mathbf{0.98 \pm 0.01}$ | ✓ |

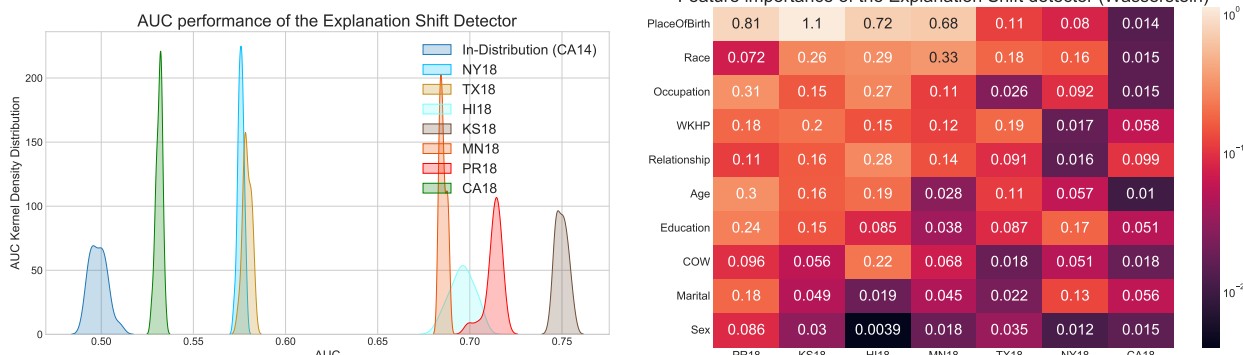

**Figure 4:** In the left figure, a comparison of the performance of *Explanation Shift Detector* in different states. In the right figure, the strength analysis of features driving the change in the model, on the y-axis are the features, and on the x-axis are the different states. Explanation shifts allow us to identify how the distribution shift of different features impacted the model.

The model $g_\psi$ is trained each time on each state using only the $\mathcal{D}_X^{new}$ in the absence of the label, and a 50/50 random train-test split evaluates its performance. As models, we use XGBoost as $f_\theta$ and logistic regression for the *Explanation Shift Detector* ($g_\psi$). The model's performance was evaluated using the AUC metric in different states, except PR18, where the model showed an explanation shift.

We hypothesize that the AUC of the Explanation Shift Detector on new data will be distinct from that on in-distribution data, primarily owing to the distinctive nature of out-of-distribution model explanations. Figure 4 illustrates the performance of our method on different data distributions, where the baseline is a ID hold-out set of CA14. The AUC for CA18, where there is only a temporal shift, is the closest to the baseline, and the OOD detection performance is better in the rest of the states. The most disparate state is Puerto Rico (PR18).

Our next objective is to identify the features where the explanations differ between $\mathcal{D}_X^{tr}$ and $\mathcal{D}_X^{new}$ data. To achieve this, we compare the distribution of linear coefficients of the detector between both distributions. We use the Wasserstein distance as a distance measure, generating 1000 in-distribution bootstraps using a 63.2% sampling fraction from California-14 and 1000 bootstraps from other states in 2018. In the right image of Figure 4, we observe that for PR18, the most crucial feature is the Place of Birth.

Furthermore, we conduct an across-task evaluation by comparing the performance of the "Explanation Shift Detector" on another prediction task in the Appendix B. Although some features are present in both prediction tasks, the weights and importance order assigned by the "Explanation Shift Detector" differ. One of this method's advantages is that it identifies differences in distributions and how they relate to the model.

## 5.5 Varying Models and Explanation Shift Detectors

OOD data detection methods based on input data distributions only depend on the detector type, independent of the model $f_\theta$. OOD Explanation methods rely on both the model and the data. Using explanation shifts as indicators for measuring distribution shifts' impact on the model enables us to account for the influencing factors of the explanation shift. Therefore, in this section, we compare the performance of different algorithms for explanation shift detection using the same experimental setup. The results of our experiments show that using Explanation Shift enables us to see differences in the choice of the original model $f_\theta$ and the Explanation Shift Detector $g_\phi$

| Detector $g_\phi$ | Estimator $f_\theta$ | | | | | | |
|---|---|---|---|---|---|---|---|
| | **XGB** | **Log.Reg** | **Lasso** | **Ridge** | **Rand.Forest** | **Dec.Tree** | **MLP** |
| **XGB** | 0.583 | 0.619 | 0.596 | 0.586 | 0.558 | 0.522 | 0.597 |
| **LogisticReg.** | 0.605 | 0.609 | 0.583 | 0.625 | 0.578 | 0.551 | 0.605 |
| **Lasso** | 0.599 | 0.572 | 0.551 | 0.595 | 0.557 | 0.541 | 0.596 |
| **Ridge** | 0.606 | 0.61 | 0.588 | 0.624 | 0.564 | 0.549 | 0.616 |
| **RandomForest** | 0.586 | 0.607 | 0.574 | 0.612 | 0.566 | 0.537 | 0.611 |
| **DecisionTree** | 0.546 | 0.56 | 0.559 | 0.569 | 0.543 | 0.52 | 0.569 |

**Table 8:** Comparison of explanation shift detection performance, measured by AUC, for different combinations of explanation shift detectors and estimators on the UCI Adult Income dataset using the Novel Covariate Group Shift experimental setup (cf. Section 5.3). The table shows that the algorithmic choice for $f_\theta$ and $g_\psi$ can impact the OOD explanation performance. We can see how, for the same detector, different $f_\theta$ models flag different OOD explanations performance. On the other side, for the same $f_\theta$ model, different detectors achieve different results.

## 5.6 Hyperparameters Sensitivity Evaluation

This section presents an extension to our experimental setup where we vary the model complexity by varying the model hyperparameters $\mathcal{S}(f_\theta, X)$. We use the UCI Adult Income dataset with the Novel Covariate Group Shift experimental setup (cf. Section 5.3). For the Stackoverflow as training data, we use the United States of America and a novel covariate group, France.

In this experiment, we changed the hyperparameters of the original model: for the decision tree, we varied the depth of the tree, while for the gradient-boosted decision trees, we changed the number of estimators, and for the random forest, both hyperparameters. We calculated the Shapley values using TreeExplainer (Lundberg et al., 2020). For the Detector choice of model, we compare Logistic Regression and XGBoost models.

The results presented in Figure 5 show the AUC of the *Explanation Shift Detector* for the ACS Income dataset under novel group shift. We observe that the distribution shift does not affect very simplistic models, such as decision trees with depths 1 or 2. However, as we increase the model complexity, the impact of out-of-distribution data on the model becomes more pronounced. Furthermore, when we compare the performance of the *Explanation Shift Detector* across different models, such as Logistic Regression and gradient-boosted decision trees, we observe distinct differences(note that the y-axis takes different values). Furthermore, in Appendix B.5, we study the effects of varying the complexity hyperparameters for the Explanation Shift Detector.

In conclusion, the explanation distributions serve as a projection of the data and model sensitive to what the model has learned. The results demonstrate the importance of considering model complexity under distribution shifts.

## 5.7 Discussion

The Shapley value, a key component in our method, describes how a model's prediction for a specific data point deviates from the mean. These theoretical considerations, which we laid out in Section 4, have been confirmed by our experimental sections.

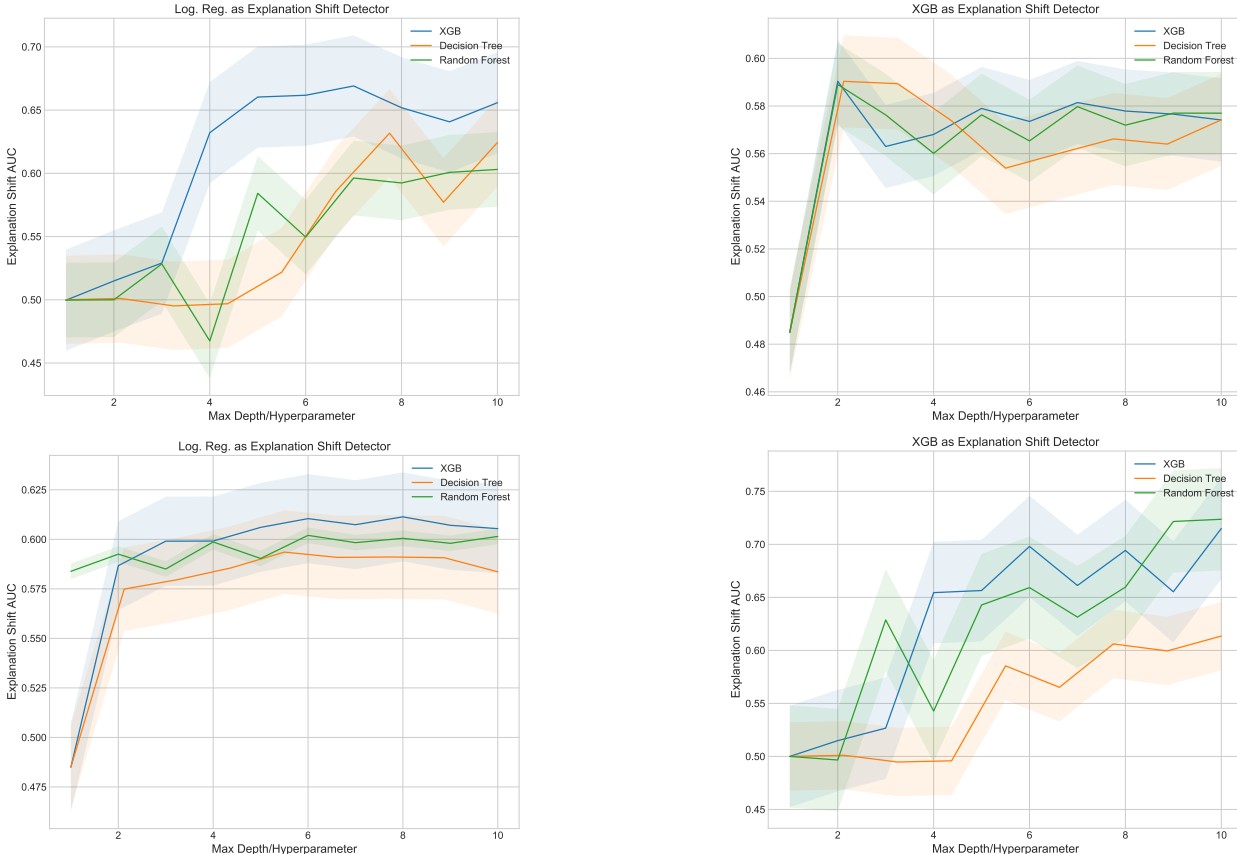

**Figure 5:** Images represent the AUC of the *Explanation Shift Detector*, on two dataset: Top ACS Income and Bottom Stackoverflow under novel group shift. In the images on the left, the detector is a logistic regression, and in the images on the right, it is a gradient-boosted decision tree classifier. By changing the model, we can see that vanilla models (decision tree with depth 1 or 2) are unaffected by the distribution shift, while when increasing the model complexity, the out-of-distribution impact of the data in the model starts to be tangible

In Section 5.3, we have studied input distribution shift. Our experiment shows that explanation shift detects input distribution shifts better than the best baseline methods. Table 7 showcases the two top-performing methods—comparing input distributions with Kolmogorov-Smirnoff *(B1)* and our method — with statistically insignificant differences.

In Section 5.2, we have studied co-variate shift. Considering, Table 6, the best method for detecting input distribution shifts, *(B1)*, fails completely on this task. The second best method is *(B2)* comparison of prediction distributions using the Wasserstein distance, which also did quite well w.r.t. predicting input distribution shifts and came rather close behind our approach in both experiments.

Moreover, in our geopolitical and temporal shift experiment (cf. Section 5.4), we demonstrate the ability to account for the drivers of model changes under such input data shifts. Cross-task comparisons in experiments (Figure 9 or Figure 8) highlight how explanation shift feature importance varies even when input distribution shifts remain constant during cross-task. These capabilities are not offered by any of the competing baselines. These observations are further supported by additional experiments in Section 5.6, where we solely vary model complexity, showcasing the adaptability of explanation shifts to changes in model characteristics.

## 6 Conclusions

Commonly, the problem of detecting the impact of the distribution shift on the model has relied on measurements for detecting shifts in the input or output data distributions or relied on assumptions either on the type of distribution shift or causal graphs availability. In this paper, we proposed explanation shifts as an indicator for detecting and identifying the impact of distribution shifts on machine learning models. We provide software, mathematical analysis examples, synthetic data, and real-data experimental evaluation. We found that measures of explanation shift can provide more insights than input distribution and prediction shift measures when monitoring machine learning models.

**Limitations:** The potential utility of explanation shifts as distribution shift indicators that affect the model in computer vision or natural language processing tasks remains an open question. We have used feature attribution explanations to derive indications of explanation shifts, but other AI explanation techniques may be applicable and come with their advantages. Also, our approach cannot detect concept shifts, as concept shift requires understanding the interaction between input data and response variables. By the nature of pure concept shifts, such changes do not affect the model. We work under the assumption that such labels are not available for new data, nor do we make other assumptions; therefore, our method is not able to predict the degradation of prediction performance under distribution shifts.

Furthermore, our use of the `shap` Python package for Shapley values approximation can introduce known drawbacks, as highlighted in recent literature (Bilodeau et al., 2022; Slack et al., 2020a). Additionally, our current implementation relies on linear Shapley value interaction approximations, which can be extended following the work of Fumagalli et al. (2023); Bordt & von Luxburg (2023).

### Reproducibility Statement

To ensure reproducibility, we make the data, code repositories, and experiments publicly available `https://github.com/cmougan/ExplanationShift`. Also, an open-source Python package `skshift`, available at: `https://skshift.readthedocs.io/`For our experiments, we used default `scikit-learn` parameters Pedregosa et al. (2011).Experiments were run on a 4 vCPU server with 32 GB RAM.

## Acknowledgements

This work has received funding from the European Union's Horizon 2020 research and innovation program under Marie Sklodowska-Curie Actions (Grant Agreement Number 860630) for the project "NoBIAS—Artificial Intelligence without Bias". The views expressed in this article are purely those of the authors and may not, under any circumstances, be regarded as an official position or policy of the European Commission. We acknowledge the support of the Stuttgart Research Focus Interchange Forum for Reflection on Intelligent Systems (IRIS).

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

# A  Experimental Comparison against Specific Related Work

## A.1  Comparison Against Changes on Feature Attribution Relevance

In this section, we present a comparative analysis against the work of (Nigenda et al., 2022),

which involves assessing the disparity in feature importance orders between training data and out-of-distribution data. To quantify this disparity, we employ the normalized discount cumulative gain (NDCG) metric. This method is versatile, accommodating both individual sample analysis and distribution-level assessments. In cases involving distributions, we aggregate the average feature importance.

### A.1.1  Novel Group Shift

**Experimental Set-Up:**This experiment extends the core experiment detailed in Section 5, where distribution shifts arise due to the emergence of previously unseen groups during the prediction phase.

**Datasets:** We use ACS Income, ASC Employment, ACS Mobility and ACS Travel time (Ding et al., 2021b). The group that is not present on the features is the *black* ethnicity.

**Baseline:** We compare against the method proposed by Nigenda et al. (2022), *(B6)* of the experimental comparison of the main body, that compares the order of the feature importance using the NDCG between train and unseen data. We vary $f_\theta$ to be a `XGBoost` and a Logistic regression. For the "Explanation Shift Detector", $g_\psi$ , we use a logistic regression in both.

**Metrics:**To facilitate a direct comparison with the Area Under the Curve (AUC) metric, we adapt the NDCG metric, to have the same interval range as follows: $(1 - NDCG) + 0.5$, ensuring a consistent metric range.

This extended experiment aims to further validate the effectiveness of the "Explanation Shift Detector" under novel group shifts in real-world datasets. It demonstrates how the approach performs consistently across

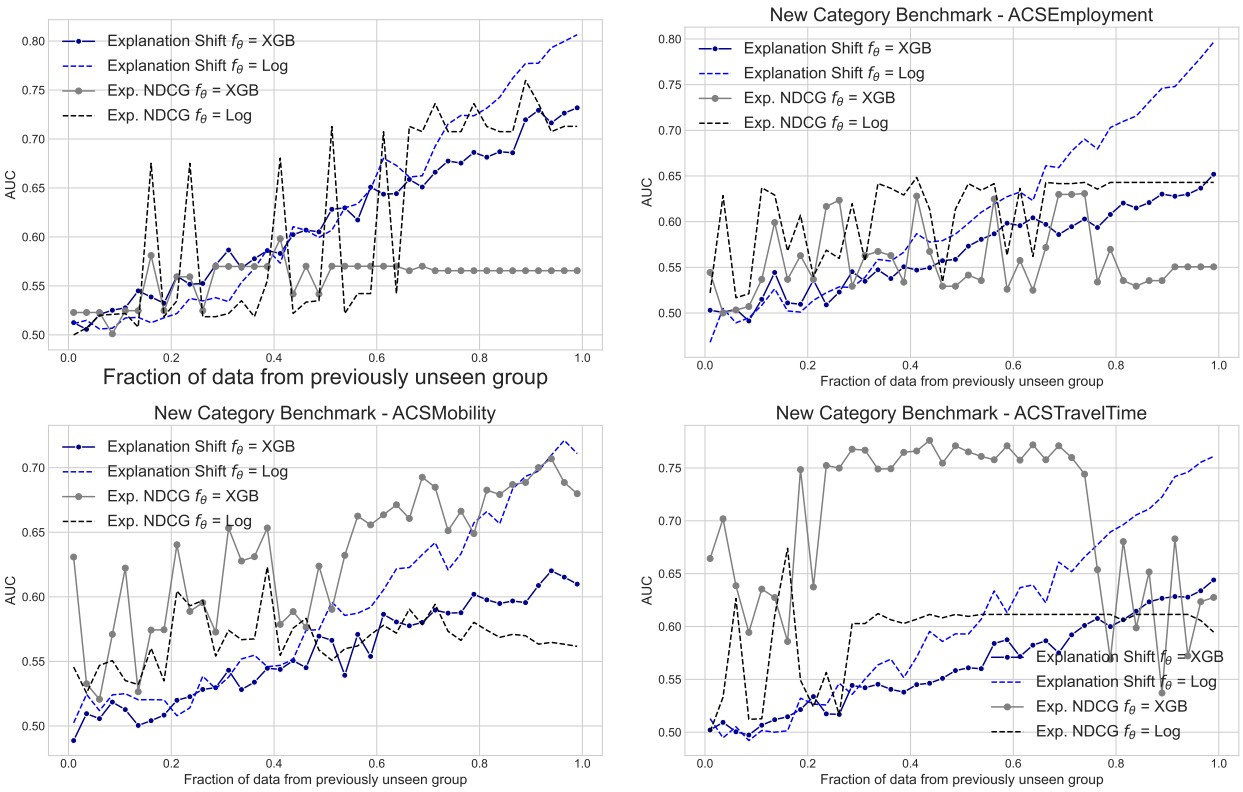

**Figure 6:** Novel group shift experiment conducted on the 4 Datasets. Sensitivity (AUC) increases as the proportion of previously unseen social groups grows. As the experimental setup has a gradual distribution shift, ideal indicators should exhibit a steadily increasing slope. However, in all figures, NDCG exhibits saturation and instability. These observations align with the analysis presented in the synthetic experiment section, as discussed in Section 5.2 of the main paper.

multiple datasets and provides insights into the sensitivity of model behavior as previously unseen social groups become a larger part of the prediction data. The results are presented in Figure 6, where our proposed method is compared against Exp. NDCG *(B6)* across the four datasets. We can see how Exp. NDCG *(B6)* is more unstable and finds often an horizontal asymptot, in all the situations, this is due to changes on the feature importance order do not have information about the value, where our approach of performing a Classifier Two Sample Test on the distributions of explanations do.

### A.1.2 Synthetic Data Comparison

In this section, we evaluate changes in the distribution of explanations and the order of feature importance when faced with a synthetic data shift scenario. We begin with a bivariate normal distribution $\mathcal{D}_X^{tr} = (X_1, X_2) \sim N(1, I)$, where $I$ represents the identity matrix of order two. We create a synthetic target variable $Y = X_1^2 \cdot X_2 + \epsilon$, and develop a machine learning model $f_\theta : \mathcal{D}_X \rightarrow \mathcal{D}_Y$ using a non-linear model, specifically an `XGBoost` model. Subsequently, we generate new data from $\mathcal{D}_X^{new} = (X_1, X_2) \sim N(2, I)$, which constitutes a shift of $D_X^{new} = D^{tr}X + 1$. We then compute SHAP values for $\mathcal{S}(f\theta, \mathcal{D}_X)$ and compare the average contributions' orders.

Having sampled $50,000$ instances from both $\mathcal{D}_X^{tr}$ and $\mathcal{D}_X^{new}$, we analyze whether alterations in explanation distributions and explanation importance orders can detect these changes. To achieve this, we compare one-tailed p-values from the Kolmogorov-Smirnov test for explanation shifts and the order of average SHAP values between the distributions.

**Table 9:** Comparison between distribution shifts in explanations and shifts in feature attribution importance orders(previous work of (Nigenda et al., 2022)). Explanation distributions exhibit differences, while the importance order remains consistent.

| Comparison | Conclusions |
|---|---|
| $\mathbf{P}(\mathcal{D}_X^{te})$, $\mathbf{P}(\mathcal{D}_X^{new})$ | Distinct |
| $\mathbf{P}(\mathcal{S}(f_\theta, \mathcal{D}_X^{te}))$, $\mathbf{P}(\mathcal{S}(f_\theta, \mathcal{D}_X^{new}))$ | Distinct |
| $\mathbf{P}(\mathcal{S}_1(f_\theta, \mathcal{D}_X^{te}) > \mathcal{S}_2(f_\theta, \mathcal{D}_X^{te}))$, $\mathbf{P}(\mathcal{S}_1(f_\theta, \mathcal{D}_X^{new}) > \mathcal{S}_2(f_\theta, \mathcal{D}_X^{new}))$ | Not Distinct |

### A.1.3 Analytical Comparison under Monotonous Uniform Shift

In this section, we conduct an analytical comparison between changes in explanation distributions and changes in the order of feature importance.

**Example A.1.** ***Comparison against NDCG*** Let $\mathcal{D}_X^{tr} = (\mathcal{D}_{X_1}^{tr}, \mathcal{D}_{X_2}^{tr}) \sim N([\mu_1, \mu_1], I)$ and $\mathcal{D}_X^{new} = (\mathcal{D}_{X_1}^{new}, \mathcal{D}_{X_2}^{new}) \sim N([\mu_2, \mu_2], I)$ where the relationship between $\mu_1$ and $\mu_2$ is monotonous uniform shift characterized by $\mu_2 = \mu_1 + N$ where N is a real number. We fit a linear model $f_\theta(X_1, X_2) = \gamma + a_1 \cdot X_1 + a_2 \cdot X_2$, where $a_1 > a_2$. Then even if the distribution of SHAP values are distinct between $\mathcal{S}(f_\theta, \mathcal{D}_X^{tr})$ and $\mathcal{S}(f_\theta, \mathcal{D}_X^{new})$, the order of importance between the distributions is not distinct. If $\mathcal{S}_1(f_\theta, \mathcal{D}_X^{tr}) > \mathcal{S}_2(f_\theta, \mathcal{D}_X^{tr})$ then $\mathcal{S}_1(f_\theta, \mathcal{D}_X^{new}) > \mathcal{S}_2(f_\theta, \mathcal{D}_X^{new})$. But the distributions are distinct $\mathcal{S}_1(f_\theta, \mathcal{D}_X^{tr}) \neq \mathcal{S}_1(f_\theta, \mathcal{D}_X^{new})$ and $\mathcal{S}_2(f_\theta, \mathcal{D}_X^{tr}) \neq \mathcal{S}_2(f_\theta, \mathcal{D}_X^{new})$.

$$\mathcal{S}_j(f_\theta, \mathcal{D}_X) = a_j \cdot (\mathcal{D}_{X_j} - \mu_1), \mathcal{S}_j(f_\theta, \mathcal{D}_X^{new}) = a_j \cdot (\mathcal{D}_{X_j}^{new} - \mu_2),$$
$$\mu_2 = \mu_1 + N.$$
$$\text{Then} \quad \mathcal{S}_j(f_\theta, \mathcal{D}_X) \neq \mathcal{S}_j(f_\theta, \mathcal{D}_X^{new}),$$
$$\text{But} \quad \mathcal{S}_1(f_\theta, \mathcal{D}_X) > \mathcal{S}_2(f_\theta, \mathcal{D}_X) \quad \Leftrightarrow \quad \mathcal{S}_1(f_\theta, \mathcal{D}_X^{new}) > \mathcal{S}_2(f_\theta, \mathcal{D}_X^{new}).$$

**Conclusion of the comparison to** Nigenda et al. (2022) In the context of natural data, when confronted with a novel covariate shift, our findings indicate that NDCG demonstrates limited sensitivity and fails to detect shifts when the fraction of data from previously unseen groups exceeds ratios 0.2 to 0.4 threshold.

Furthermore, in our analyses both synthetic and natural data, we observe that NDCG struggles to provide accurate and consistent estimates when faced with multicovariate shifts.

Both analytically and in our experiments with synthetic data, it becomes evident that NDCG lacks robustness and sensitivity when confronted with even a basic, uniform, and monotonous shift.

## B  Further Experiments on Real Data

In this section, we extend the prediction task of the main body of the paper. The methodology used follows the same structure. We start by creating a distribution shift by training the model $f_\theta$ in California in 2014 and evaluating it in the rest of the states in 2018, creating a geopolitical and temporal shift. The model $g_\theta$ is trained each time on each state using only the $X^{New}$ in the absence of the label, and a 50/50 random train-test split evaluates its performance. As models, we use a gradient-boosted decision tree(Chen & Guestrin, 2016; Prokhorenkova et al., 2018) for $f_\theta$, approximating the Shapley values by TreeExplainer (Lundberg et al., 2020), and using logistic regression for the *Explanation Shift Detector*.

### B.1  ACS Employment

The objective of this task is to determine whether an individual aged between 16 and 90 years is employed or not. The model's performance was evaluated using the AUC metric in different states, except PR18, where the model showed an explanation shift. The explanation shift was observed to be influenced by features such as Citizenship and Military Service. The performance of the model was found to be consistent across most

of the states, with an AUC below 0.60. The impact of features such as difficulties in hearing or seeing was negligible in the distribution shift impact on the model. The left figure in Figure 7 compares the performance of the Explanation Shift Detector in different states for the ACS Employment dataset.

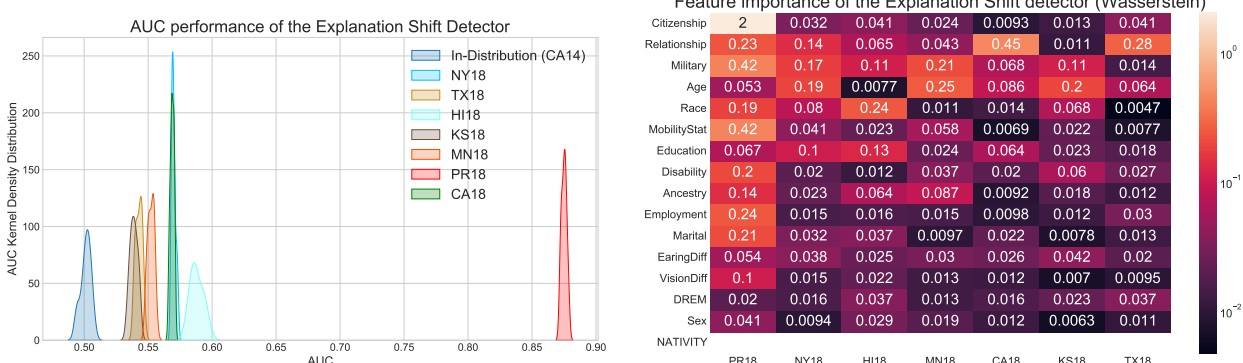

**Figure 7:** The left figure compares the performance of the Explanation Shift Detector in different states for the ACS Employment dataset. The right figure shows the feature importance analysis for the same dataset.

Additionally, the feature importance analysis for the same dataset is presented in the right figure in Figure 7.

## B.2 ACS Travel Time

The goal of this task is to predict whether an individual has a commute to work that is longer than +20 minutes. For this prediction task, the results differ from the previous two cases; the state with the highest OOD score is $KS18$, with the "Explanation Shift Detector" highlighting features such as Place of Birth, Race or Working Hours Per Week. The closest state to ID is CA18, where there is only a temporal shift without any geospatial distribution shift.

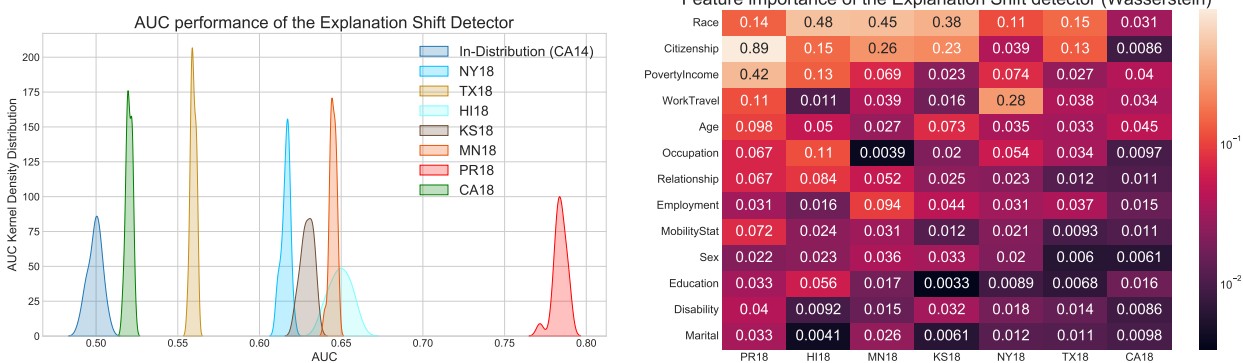

**Figure 8:** In the left figure, comparison of the performance of *Explanation Shift Detector*, in different states for the ACS TravelTime prediction task. In the left figure, we can see how the state with the highest OOD AUC detection is KS18 and not PR18 as in other prediction tasks; this difference concerning the other prediction task can be attributed to "Place of Birth", whose feature attributions the model finds to be more different than in CA14.

## B.3 ACS Mobility

The objective of this task is to predict whether an individual between the ages of 18 and 35 had the same residential address as a year ago. This filtering is intended to increase the difficulty of the prediction task, as the base rate for staying at the same address is above 90% for the population (Ding et al., 2021b).

The experiment shows a similar pattern to the ACS Income prediction task (cf. Section 4), where the inland US states have an AUC range of $0.55 - 0.70$, while the state of PR18 achieves a higher AUC. For PR18, the model has shifted due to features such as Citizenship, while for the other states, it is Ancestry (Census record of your ancestors' lives with details like where they lived, who they lived with, and what they did for a living) that drives the change in the model.

As depicted in Figure 9, all states, except for PR18, fall below an AUC of explanation shift detection of 0.70. Protected social attributes, such as Race or Marital status, play an essential role for these states, whereas for PR18, Citizenship is a key feature driving the impact of distribution shift in the model.

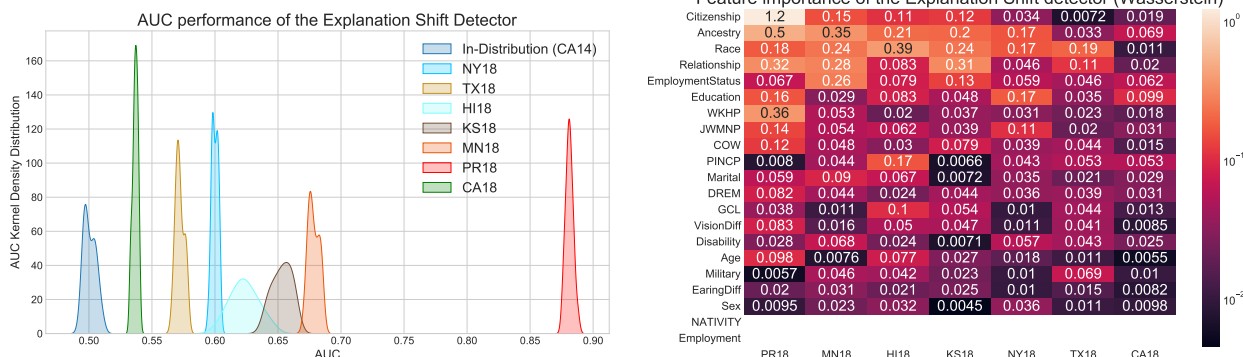

**Figure 9:** Left figure shows a comparison of the *Explanation Shift Detector*'s performance in different states for the ACS Mobility dataset. Except for PR18, all other states fall below an AUC of explanation shift detection of 0.70. The features driving this difference are Citizenship and Ancestry relationships. For the other states, protected social attributes, such as Race or Marital status, play an important role.

## B.4    StackOverflow Survey Data: Novel Covariate Group

This experimental section evaluates the proposed Explanation Shift Detector approach on real-world data under novel group distribution shifts. In this scenario, a new unseen group appears at the prediction stage, and the ratio of the presence of this unseen group in the new data is varied. As a training data country, we use the United States. The model $f_\theta$ used is a gradient-boosted decision tree or logistic regression, and logistic regression is used for the detector. The results show that the AUC of the Explanation Shift Detector varies depending on the quantification of OOD explanations, and it shows more sensitivity concerning model variations than other state-of-the-art techniques.

The dataset used is the StackOverflow annual developer survey, with over 70,000 responses from over 180 countries examining aspects of the developer experience (Stackoverflow, 2019). The data has high dimensionality, leaving it with +100 features after data cleansing and feature engineering. The goal of this task is to predict the total annual compensation.

## B.5    Sensitivity Analysis of Explanation Shift Detector Hyperparameters

This section extends our experimental setup by evaluating the impact of varying the complexity of the Explanation Shift Detector through its hyperparameters, represented as $g\psi(\mathcal{S}(f\theta, X))$. We focus on the geopolitical and temporal shifts in the US Income dataset, using CA14 for training and PR18 as the out-of-distribution (OOD) dataset.

In this analysis, we adjusted the hyperparameters for the Explanation Shift Detector across different model types. For decision trees, we varied tree depth; for gradient-boosted decision trees, we changed the number of estimators; and for random forests, both hyperparameters were adjusted. To evaluate estimator choices, we compared Logistic Regression and `XGBoost` models. While this setup resembles the one in Section 5.6, here we focus on altering the complexity of the Explanation Shift Detector rather than the original estimator.

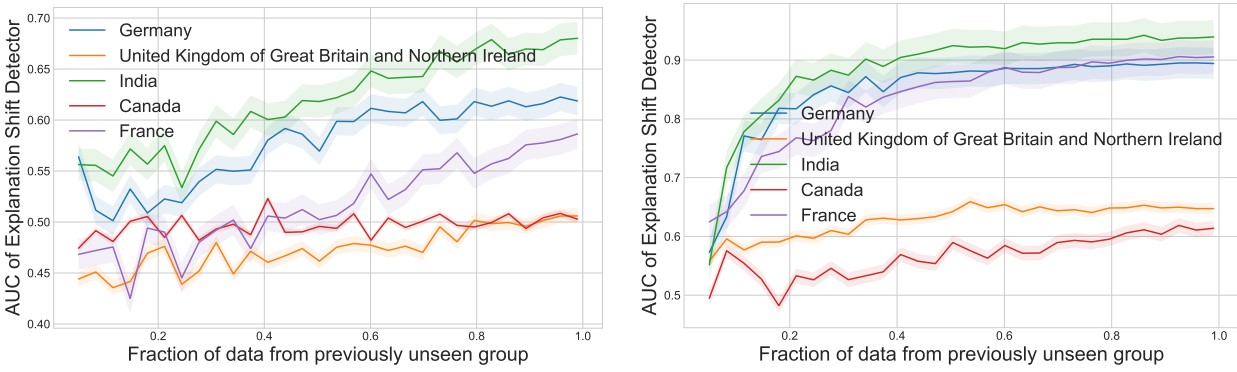

**Figure 10:** Both images represent the AUC of the *Explanation Shift Detector* for different countries on the StackOverflow survey dataset under novel group shift. In the left image, the estimator, $f_\theta$, is a gradient-boosted decision tree; in the right image, for both cases the detector, $g_\psi$, is a logistic regression. By changing the type of estimator model, we can see how different types of models are affected differently for the same distribution shift.

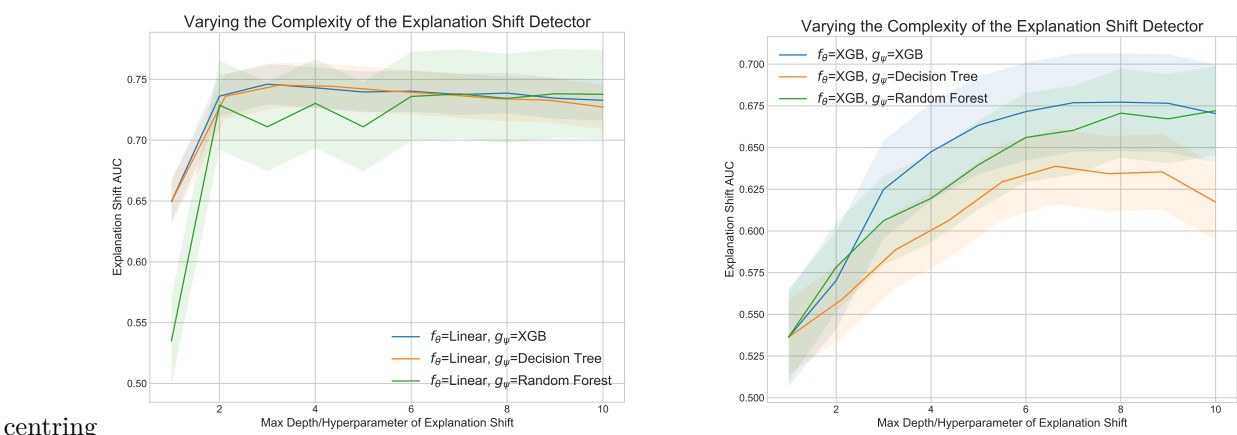

centring

**Figure 11:** The AUC of the *Explanation Shift Detector* under geopolitical and temporal shifts in the ACS Income dataset. Left: results for a linear model as the estimator. Right: results for a gradient-boosted decision tree regressor. Increasing the complexity of the Explanation Shift Detector highlights the impact of distribution shifts, particularly for the XGB model. For linear models, the AUC plateaus due to their limited complexity.

As shown in Figure 11, the Explanation Shift Detector requires a minimum level of complexity to detect the effects of distribution shifts. Beyond this threshold, the AUC values plateau, indicating diminishing returns from increasing the Explanation Shift Detector's complexity. For the linear model as estimators ($f_\theta$), the AUC of the Explanation Shift Detector plateaus with less complex parameter, likely due to the simplicity of how the distribution shift impact a simpler model. In contrast, when the XGB model increases the detector's complexity, it continues to improve performance, reflecting the model's greater capacity to learn complex patterns.

In conclusion, explanation distributions act as projections of the data and model behavior, capturing the impact of learned features. While Explanation Shift Detector complexity is important under distribution shifts, its influence appears secondary to that of the original estimator model complexity, as explored in Section 5.6.

## C   LIME as an Alternative Explanation Method

Another feature attribution technique that satisfies the properties above (efficiency and uninformative features Section 2) and can be used to create the explanation distributions is LIME (Local Interpretable Model-

Agnostic Explanations). The intuition behind LIME is to create a local interpretable model that approximates the behavior of the original model in a small neighbourhood of the desired data to explain (Ribeiro et al., 2016b;a) whose mathematical intuition is very similar to the Taylor series. In this work, we have proposed explanation shifts as a key indicator for investigating the impact of distribution shifts on ML models. In this section, we compare the explanation distributions composed by SHAP and LIME methods. LIME can potentially suffer several drawbacks:

**Computationally Expensive:** Its current implementation is more computationally expensive than current SHAP implementations such as TreeSHAP (Lundberg et al., 2020), Data SHAP (Kwon et al., 2021; Ghorbani & Zou, 2019) or Local and Connected SHAP (Chen et al., 2019), the problem increases when we produce explanations of distributions. Even though implementations might be improved, LIME requires sampling data and fitting a linear model, which is a computationally more expensive approach than the aforementioned model-specific approaches to SHAP.

**Local Neighborhood:** The definition of a local "neighborhood", which can lead to instability of the explanations. Slight variations of this explanation hyperparameter lead to different local explanations. In Slack et al. (2020b) the authors showed that the explanations of two very close points can vary greatly.

**Dimensionality:** LIME requires as a hyperparameter the number of features to use for the local linear approximation. This creates a dimensionality problem as for our method to work, the explanation distributions must be from the exact same dimensions as the input data. Reducing the number of features to be explained might improve the computational burden.

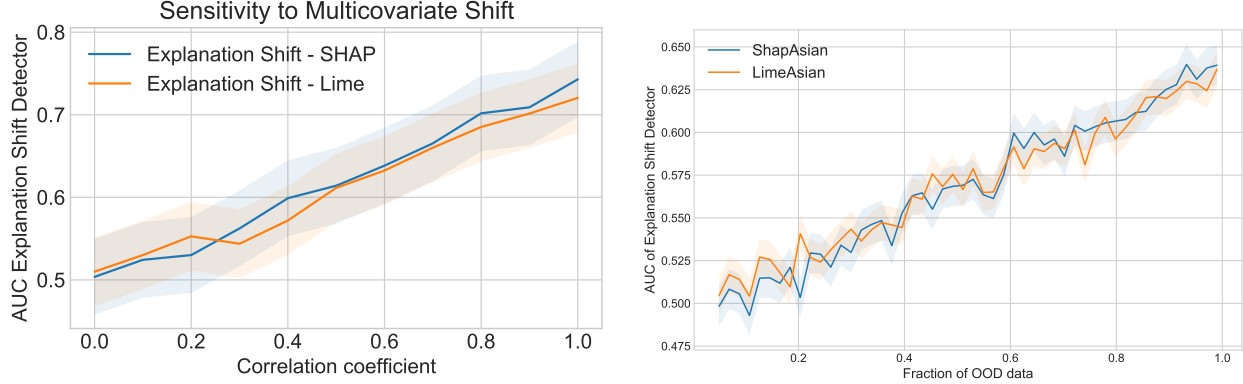

**Figure 12:** Comparison of the explanation distribution generated by LIME and SHAP. The left plot shows the sensitivity of the predicted probabilities to multicovariate changes using the synthetic data experimental setup of 2 on the main body of the paper. The right plot shows the distribution of explanation shifts for a New Covariate Category shift (Asian) in the ASC Income dataset.

Figure 12 compares the explanation distributions generated by LIME and SHAP. The left plot shows the sensitivity of the predicted probabilities to multicovariate changes using the synthetic data experimental setup from Figure 2 in the main body of the paper. The right plot shows the distribution of explanation shifts for a New Covariate Category shift (Asian) in the ASC Income dataset. The performance of OOD explanations detection is similar between the two methods. Still, LIME suffers from two drawbacks: its theoretical properties rely on the definition of a local neighborhood, which can lead to unstable explanations (false positives or false negatives on explanation shift detection), and its computational runtime required is much higher than that of SHAP (see experiments below).

## C.1 Runtime

We analyzed the runtimes of generating the explanation distributions using the two proposed methods. The experiments were run on a server with four vCPUs and 32 GB of RAM. We used `shap` version 0.41.0 and `lime` version 0.2.0.1 as software packages. In order to define the local neighborhood for both methods, we use all the data provided as background data in this example. As an $f_\theta$ model, we use an `XGBoost` and compare

the results of TreeShap against LIME. When varying the number of samples we use five features and while varying the number of features we use 1000 samples.

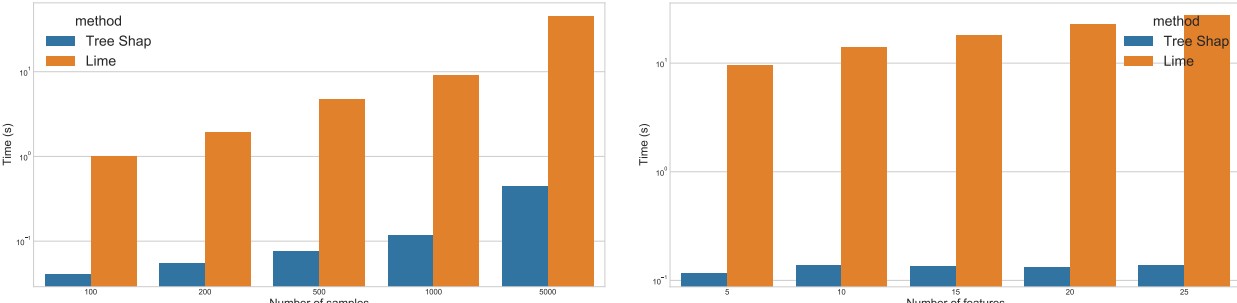

**Figure 13:** Wall time for generating explanation distributions using SHAP and LIME with different numbers of samples (left) and different numbers of columns (right). Note that the y-scale is logarithmic. The experiments were run on a server with four vCPUs and 32 GB of RAM. The runtime required to create an explanation distribution with LIME is far greater than SHAP for a gradient-boosted decision tree

Figure 13, shows the wall time required to generate explanation distributions using SHAP and LIME with varying samples and columns. The runtime required of generating an explanation distributions using LIME is much higher than using SHAP, mainly when producing explanations for distributions. This is because LIME requires training a local model for each instance of the input data to be explained, which can be computationally expensive. In contrast, SHAP relies on heuristic approximations to estimate the feature attribution without training a model for each instance. The results illustrate that this difference in computational runtime becomes more pronounced as the number of samples and columns increases.

We note that limiting the number of features to be explained can further reduce the computational burden of generating the explanation distributions, as this reduces the dimensionality of the explanation distributions. However, this will inhibit the quality of the explanation shift detection, as it won't be able to detect changes in the distribution shift that impact the model on those features.

Given the current state-of-the-art software packages, we have used SHAP values due to the lower runtime required and the theoretical guarantees that hold with the implementations. In the experiments performed in this paper, we are dealing with a medium-scaled dataset with around $\sim 1,000,000$ samples and $20-25$ features. Further work can be envisioned on developing novel mathematical analysis and software that study under which conditions which method is more suitable.

# D  True to the Model or True to the Data?

The "Explanation Shift Detector" proposed in this work relies on the explanation distributions that satisfy efficiency and uninformative theoretical properties. We have used the Shapley values as an explainable AI method that satisfies these properties. However, the correct way to connect a model to a coalitional game, which is the central concept of Shapley values, is a source of controversy, with two main approaches (*i*) an interventional (Aas et al., 2021; Frye et al., 2020; Zern et al., 2023) or (*ii*) an observational formulation of the conditional expectation(Sundararajan & Najmi, 2020).

In the following experiment, we compare what are the differences between estimating the Shapley values using one or the other approach. We benchmark this experiment on the four prediction tasks based on the US census data (Ding et al., 2021a) and using the "Explanation Shift Detector", where both the model $f_\theta(X)$ and $g_\psi(\mathcal{S}(f_\theta, X))$ are linear models. We will calculate the Shapley values using the SHAP linear explainer. [2]

The comparison depends on a feature perturbation hyperparameter: whether the approach to compute the SHAP values is either *interventional* or *correlation dependent*. The interventional SHAP values break the dependence structure between features in the model to uncover how the model would behave if the inputs

---

[2]`https://shap.readthedocs.io/en/latest/generated/shap.explainers.Linear.html`

were changed (as it was an intervention). This option is said to stay "true to the model", meaning it will only give allocation credit to the features that the model actually uses (Aas et al., 2021).

On the other hand, the full conditional approximation of the SHAP values respects the correlations of the input features. If the model depends on one input that is correlated with another input, then both get some credit for the model's behaviour. This option is said to say "true to the data", meaning that it only considers how the model would behave when respecting the correlations in the input data (Chen et al., 2020).In our case, we will measure the difference between the two approaches by looking at the linear coefficients of the model $g_\psi$ and comparing the performance using the geo-political and temporal experiment of the previous section 5, for this case between CA14 and PR18.

**Table 10:** AUC comparison of the "Explanation Shift Detector" between estimating the Shapley values between the interventional and the correlation-dependent approaches for the four prediction tasks based on the US census dataset (Ding et al., 2021a). The % character represents the relative difference. The performance differences are negligible.

|  | Interventional | Observational | % |
|---|---|---|---|
| Income | 0.736438 | 0.736439 | 1.1e-06 |
| Employment | 0.747923 | 0.747923 | 4.44e-07 |
| Mobility | 0.690734 | 0.690735 | 8.2e-07 |
| Travel Time | 0.790512 | 0.790512 | 3.0e-07 |

**Table 11:** Linear regression coefficients comparison of the "Explanation Shift Detector" between estimating the Shapley values between the interventional and the correlation-dependent approaches for one of the US census-based prediction tasks (ACS Income). The % character represents the relative difference. The coefficients show negligible differences between the calculation methods

|  | Interventional | Observational | % |
|---|---|---|---|
| Marital | 0.348170 | 0.348190 | 2.0e-05 |
| Worked Hours | 0.103258 | -0.103254 | 3.5e-06 |
| Class of worker | 0.579126 | 0.579119 | 6.6e-06 |
| Sex | 0.003494 | 0.003497 | 3.4e-06 |
| Occupation | 0.195736 | 0.195744 | 8.2e-06 |
| Age | -0.018958 | -0.018954 | 4.2e-06 |
| Education | -0.006840 | -0.006840 | 5.9e-07 |
| Relationship | 0.034209 | 0.034212 | 2.5e-06 |

In Table 10 and Table 11, we can see the comparison of the effects of using the aforementioned approaches to learn our proposed method, the "Explanation Shift Detector". Even though the two approaches differ theoretically, the differences become negligible when explaining the protected characteristic, i.e. when providing the linear regression coefficients.

