# OpenReview forum: "Explanation Shift: How Did the Distribution Shift Impact the Model?"
_TMLR — Accepted by TMLR_

### Review · Reviewer_NeeN · 2024-09-08

**Summary Of Contributions:**

The authors introduce a new definition of explanation shift as the statistical comparison between how train data and new data contribute/ related to predictions measured by shapely values. The author further introduces the pipeline along with metrics to measure distribution and explanation shift. Experiments were done on synthetic and real tabular data to demonstrate the pipeline.

**Audience:**

Yes

**Broader Impact Concerns:**

This is well discussed in the discussion section.

**Claims And Evidence:**

Yes

**Requested Changes:**

- When I was reading definition 3.2, I got upset with the implicit assumption that there is no overfitting/underfitting for the model f$\theta$. Though this is discussed in the later paragraph, I think that all of the assumptions should be clearly stated.

**Strengths And Weaknesses:**

Overall, I am very happy with the work, and I do not have many criticisms toward it.

Strength:
- The demonstration of the pipeline in terms of theories (multivariate shifts and uninformative shift) and empirical results are comprehensive enough

- The effectiveness of the methodology is good in terms of sensitivities

Weakness:

- The pipeline relies on a trained classifier to predict if an explanation should be classified as ID or OOD. This would rely on a well trained classifier as a prerequisite to ensure the pipeline works properly. Though it is followed by roc and two sample test, which could lower the risk of having a weakly trained classifier, the risk would be concerning if applied to more complicated data such as images or texts.

- Perhaps adding some discussions or ablations about the on how the choices and the training quality (well trained or not) of the classifier would impact the result would be good. (Since as discussed in the discussion section that the extension to cv and nlp is still under developing, I agree that it would not be the ultimate focus of this paper. )

---

> ### Author Response · Authors · 2024-10-27
>
> Dear reviewer,
>
> Many thanks for the comments!
>
> > When I was reading definition 3.2, I got upset with the implicit assumption that there is no overfitting/underfitting for the model f
> . Though this is discussed in the later paragraph, I think that all of the assumptions should be clearly stated.
>
> In Section 5.6, "Hyperparameter Sensitivity Evaluation," we examine how varying $\theta$ affects explanation shift. One finding from this section is that underfitted models are indeed less susceptible to distribution shift, whereas more complex models tend to be more sensitive. We agree that this dependency on $\theta$ in relation to explanation shifts could be emphasized earlier in the paper to clarify assumptions up front.
>
> Thank you also for agreeing that extending to CV and NLP domains is beyond the primary scope of this paper.

---

> > ### Comment · Reviewer_NeeN · 2024-10-31
> >
> > Thank you for your response!
> >
> > > One finding from this section is that underfitted models are indeed less susceptible to distribution shift, whereas more complex models tend to be more sensitive. We agree that this dependency on
> >  in relation to explanation shifts could be emphasized earlier in the paper to clarify assumptions up front.
> >
> > Please include this (maybe just one or two short descriptions in the earlier paragraphs) in the newer version. I believe this would help to clarify the paper a lot.
> >
> > Other than that, my concerns are all addressed.

---

### Review · Reviewer_Crub · 2024-10-11

**Summary Of Contributions:**

This paper introduces the concept of **explanation shift**, defined as a statistical comparison between how model predictions are explained on training data versus target data. The approach involves an **explanation function** $ \mathcal{S}(f_{\theta}, x) $, which takes the model $ f_{\theta} $ and a $ p $-dimensional data point $ x \in \mathbb{R}^{p} $, and returns a vector of attributions $ \mathcal{S}(f_{\theta}, x) \in \mathbb{R}^{p} $.

Initially, this function is fit on the validation set of the source domain. It is then re-applied to the test set of the target domain to detect differences in feature attributions between the two domains. A classifier is trained to distinguish between source and target domain attributions. The classifier’s AUROC is subsequently calculated, and a two-sample classifier test (utilizing one-tailed p-values from the Kolmogorov-Smirnov test) is performed to determine whether the AUROC significantly deviates from 0.5, indicating the presence of a distribution shift.

The authors compare their explanation shift-based monitoring method with other approaches focused on different types of distribution shifts. The findings suggest that monitoring explanation shifts provides a more sensitive indicator for detecting changes in model behavior.

**Audience:**

Yes

**Broader Impact Concerns:**

There are no broader impact concerns if the requested changes are appropriately addressed.

**Claims And Evidence:**

Yes

**Requested Changes:**

It would be nice to answer the points pointed out in the above Weaknesses, reflect the answers to correct the parts that need to be modified in the text of the text, and if there are additional experiments that are conducted, it would be good to include them in the text.

**Writing Suggestions**

- The paper feels like it hasn't been thoroughly proofread. The current quality falls short of TMLR's publication standards, and substantial corrections are required throughout the manuscript.
- **Example 4.1–4.3**: The equations are understandable but need better alignment and formatting. They are currently ill-posed and visually unclear.
- **Example 4.2**: The formulas below Example 4.2 appear strange—are they directly related to Example 4.2?
- **Avoid Redundancy**: Repeating information multiple times should be avoided. For instance, the table caption text like “Displayed results are the one-tailed p-values of the Kolmogorov-Smirnov test comparison between two underlying distributions” or “Novel Covariate Group Shift for the 'Asian' group with a fraction ratio of 0.5 as described in Section 5” only needs to be mentioned once in the main text.
- **Gradient boosting decision tree** / **gradient-boosting decision tree** ⇒ Should be consistently referred to as **gradient-boosted decision tree**.

**Typos and Formatting Errors**

- (p. 2) `{(xtr_0 , ytr_0) . . . , (xtr_n , ytr_n)}` ⇒ `{(xtr_0 , ytr_0), . . . , (xtr_n , ytr_n)}`
- (p. 5) `$R^p \rightarrow {0,1}$` ⇒ `$\mathbb{R}^{p} \rightarrow {0, 1}$`
- (p. 10) `the Appendix in sectionA` ⇒ `the Appendix in Section A`
- (p. 10) `paper(cf. Section 4):` ⇒ `paper (cf. Section 4):`
- (p. 10) `shift(cf. Example 4.1 )a` ⇒ `shift (cf. Example 4.1 ) a`
- (p. 10) `shift(cf. Example 4.2) .` ⇒ `shift (cf. Example 4.2).`
- (p. 11) `datasests` ⇒ `datasets`
- (p. 15) `Section› 5.3,` ⇒ `Section 5.3`
- (p. 15) `wrt.` ⇒ `w.r.t`

**Strengths And Weaknesses:**

**Strengths**
- The concept of explanation shift—measuring changes in feature attributions across domains to detect distribution shifts—is both novel and intriguing.
- The paper provides several case studies in Section 4, aiding in the understanding of the proposed method and its ability to detect various distribution shifts. In Section 5, the authors offer experimental evidence demonstrating improved performance in detecting distribution shifts.
- Additionally, the source code for the experiments is made publicly available. The authors also contribute an open-source Python package, **skshift**, which implements the "Explanation Shift Detector" and includes tutorials to facilitate reproducibility.

**Weaknesses**
1. **Lack of Detailed Behavioral Analysis**:
   While the method can detect whether a distribution shift has occurred and the model’s behavior has changed, it does not explain *how* the model's behavior has changed or how feature attributions have shifted. This lack of interpretability limits the method's applicability. The authors should address these concerns, specifically how feature attribution changes and what these changes mean in practice.

2. **Concept Shift Detection**:
   While it is promising that the method can detect concept shift when test labels are available, this scenario is often impractical. The authors should explore whether it can detect concept shift with only a few available test labels.

3. **Modality Scope**:
   The proposed explanation shift method appears modality-agnostic, and while the authors acknowledge this in the limitations section, it raises the question of why the scope of the application was limited to the tabular domain. It would be beneficial to explore whether promising results could also be achieved in vision or NLP domains.

4. **Use of SHAP for Feature Attribution**:
   The method uses SHAP, but the authors mention known drawbacks of SHAP in the limitations section. Could better performance be achieved using more advanced feature attribution methods? The authors should address these concerns and explain whether those drawbacks could be mitigated with other techniques.

5. **Classifier Choice**:
   Why didn’t the authors explore using more complex classifiers, like deep neural networks, instead of logistic regression? Is this decision driven by the characteristics of the tabular domain?

6. **Synthetic Data in Experiments**:
   A significant portion of the analysis and experiments was conducted on synthetic data. In particular, the claim in Table 5 that the explanation shift detector can handle *all* covariate shifts and uninformative shifts seems overgeneralized. The authors should provide more robust guarantees.

7. **Limited Experimental Scope**:
   Modern work in the tabular domain often involves testing on dozens of datasets to prove performance improvements [1]. The experimental scope feels too limited, especially since the main experiments were conducted on the ACS Income dataset, with only additional tests on ACS Travel Time, ACS Employment, and StackOverflow. This raises concerns about potential cherry-picking of datasets.

8. **Limited Performance Improvement**:
   In Tables 6 and 7, the performance improvement is marginal when compared to methods like B2 (Prediction Wasserstein), B7 (C2ST Output), and B1 (Input KS).

9. **Computational Cost**:
   The computational cost of the proposed method is likely very high, especially since SHAP must be calculated for every dataset. This raises concerns about scalability, particularly for larger datasets. An analysis of computational cost, comparing it with other baselines, is necessary.

[1] Yan et al. "Making Pre-trained Language Models Great on Tabular Prediction." ICLR 2024.

---

> ### Author Response · Authors · 2024-10-26
> **Thanks & Rebuttal**
>
> Dear reviewer,
>
> Many thanks for the high-quality review. Its very appreciated!
> We now proceed to the rebuttal.
>
> > Lack of Detailed Behavioral Analysis: While the method can detect whether a distribution shift has occurred and the model’s behavior has changed, it does not explain how the model's behavior has changed or how feature attributions have shifted. This lack of interpretability limits the method's applicability. The authors should address these concerns, specifically how feature attribution changes and what these changes mean in practice.
>
> We believe our method directly addresses this concern by specifically focusing on "explaining how the model's behavior has changed or how feature attributions have shifted." This is demonstrated through model $g$, as seen in Figure 1, particularly in the final blue box, "Explaining Explanation Shift Detector." Additionally, Figure 4b provides a concrete example with real data showing how feature attributions shift over time. The paper further supports these findings with both synthetic and real data experiments, alongside mathematical analysis, to illustrate how feature attributions evolve under varying conditions.
>
> Could the reviewer please clarify if there are additional concerns we might address, or if there is an aspect of the explanation that we might expand upon?
>
>
> > Concept Shift Detection: While it is promising that the method can detect concept shift when test labels are available, this scenario is often impractical. The authors should explore whether it can detect concept shift with only a few available test labels.
>
> We agree that the need for test labels to detect concept shifts can be challenging in real-world applications. However, without any labelled test data, accurately detecting concept shifts becomes infeasible. While semi-supervised methods have indeed evolved to address similar scenarios after the development of supervised approaches, adapting our method to function with limited labels represents a distinct problem setting that would require a substantial new methodology and would likely warrant a dedicated paper. Nonetheless, we appreciate the suggestion as a potential avenue for future work and agree it is an important consideration for advancing this area.
>
> > Modality Scope: The proposed explanation shift method appears modality-agnostic, and while the authors acknowledge this in the limitations section, it raises the question of why the scope of the application was limited to the tabular domain. It would be beneficial to explore whether promising results could also be achieved in vision or NLP domains.
>
> We appreciate the reviewer’s interest in extending the explanation shift method to other domains. Our method leverages the structured, fixed matrix format unique to tabular data, a feature that enables our current approach. Extending it to more complex data types, such as vision or NLP, would require significant methodological adaptations, which fall beyond the scope of this paper. Instead, our goal was to conduct a thorough and detailed analysis within the tabular modality, as reflected in our manuscript's length (~30 pages). We believe this depth of focus provides a strong foundation for future exploration in other modalities.
>
> > Use of SHAP for Feature Attribution: The method uses SHAP, but the authors mention known drawbacks of SHAP in the limitations section. Could better performance be achieved using more advanced feature attribution methods? The authors should address these concerns and explain whether those drawbacks could be mitigated with other techniques.
>
>  We acknowledge the reviewer's interest in alternative feature attribution methods and have indeed explored several approaches:
>  - in Appendix E, we analyze different SHAP estimation methods (interventional vs. observational)
>  - in Appendix D, we compare SHAP with LIME as an alternative.
>
> Each of these analyses includes a comparison of their benefits and limitations within our framework. If the reviewer has specific concerns or additional feature attribution methods in mind, we would be open to considering those as well.
>
> > Classifier Choice: Why didn’t the authors explore using more complex classifiers, like deep neural networks, instead of logistic regression? Is this decision driven by the characteristics of the tabular domain?
>
> In Section 5.5, "Varying Models and Explanation Shift Detectors," we explored a variety of classifier combinations for both
> $f$ and $g$, including Gradient Boosting Decision Trees, Random Forest, Generalized Linear Models, and Multi-Layer Perceptrons.
>
> We chose not to incorporate more complex neural network architectures, as these often do not yield optimal performance in the tabular data domain[1]
>
> [1] Why do tree-based models still outperform deep learning on typical tabular data? NeurIPS 2022

---

> > ### Author Response · Authors · 2024-10-26
> >
> > > Synthetic Data in Experiments: A significant portion of the analysis and experiments was conducted on synthetic data. In particular, the claim in Table 5 that the explanation shift detector can handle all covariate shifts and uninformative shifts seems overgeneralized. The authors should provide more robust guarantees.
> >
> > We appreciate the reviewer’s concern regarding the use of synthetic data. In the mathematical analysis section, we provide more formal guarantees for specific types of distribution shift, with these guarantees only holding for exact SHAP value calculations, achievable only under conditions of linear models and IID data. We have framed more formally Proposition 1 (Section 4.2), that does not depend on these specific conditions.
> >
> >  Additionally, while we explored different SHAP estimations and included LIME for comparison, [it’s worth noting that theoretical guarantees do not strictly apply to LIME or alternative SHAP approximations]; however, the experimental results were not impacted.
> >
> > > Limited Experimental Scope: Modern work in the tabular domain often involves testing on dozens of datasets to prove performance improvements [1]. The experimental scope feels too limited, especially since the main experiments were conducted on the ACS Income dataset, with only additional tests on ACS Travel Time, ACS Employment, and StackOverflow. This raises concerns about potential cherry-picking of datasets.
> >
> > We acknowledge the reviewer's point about dataset diversity, which is often beneficial for large-scale performance benchmarking in the tabular domain. However, our research question centers on how explanation shifts occur, making our approach more qualitative than quantitative. Extensive benchmarking across many datasets might be less insightful given this focus. To support further experimental exploration, we have provided a Python package, "skshift," which includes detailed tutorials on applying our method, including on the "Breast Cancer" dataset. This package follows a scikit-learn API structure, making it easily extendable for additional datasets as needed.
> >
> > If the reviewer still considers it we can the experiments of the Python package in the paper appendix.
> >
> > > Limited Performance Improvement: In Tables 6 and 7, the performance improvement is marginal when compared to methods like B2 (Prediction Wasserstein), B7 (C2ST Output), and B1 (Input KS).
> >
> > One may note that ours is the only approach that achieves superior behaviour in both evaluations against all approaches. The only approach that comes close in Table 7 (b1), shows very poor behaviour in Table 6 and vice versa. We suggest integrating the two tables into one table to make this large improvement by our approach more visible.
> >
> > > Computational Cost: The computational cost of the proposed method is likely very high, especially since SHAP must be calculated for every dataset. This raises concerns about scalability, particularly for larger datasets. An analysis of computational cost, comparing it with other baselines, is necessary.
> >
> > In Appendix D, particularly Figure 12 is a wall time experiment comparing TreeSHAP vs LIME. The results show that SHAP is not so expensive, given that the hardware for this paper is 4 vCPUs and 32 GB of RAM.(particularly compared to other deep learning approaches)
> >
> >
> > With respect to the writing suggestions and minor typos. Many thanks! Those will be updated and integrated!

---

> > > ### Comment · Reviewer_Crub · 2024-11-04
> > > **Official Comment by Reviewer Crub**
> > >
> > > I appreciate the detailed explanations provided in the rebuttal. Most of my concerns have been satisfactorily addressed. Thank you for your thorough response!

---

### Review · Reviewer_6Tsi · 2024-10-23

**Summary Of Contributions:**

The paper proposes an approach for detecting the impact of the distribution shift on the model performance. The authors focus on detecting shifts that affect the behaviour of a particular model trained on labelled source data and evaluated over an unlabelled target domain. The approach relies on the computation of Shapley values using a pre-fixed set of features in the training data and predicted features on the target data. The authors evaluate their approach over a variety of baselines using both synthetic and real data, revealing the greater sensitivity of explanation shifts towards model performance compared to data shifts.

**Audience:**

Yes

**Broader Impact Concerns:**

There isn't any immediate concerns in this aspect.

**Claims And Evidence:**

Yes

**Requested Changes:**

The changes requested are related to the weaknesses stated above and have been told elaborately. Here they are reiterated briefly for completeness. But refer to the points above in the 'weaknesses' section above for detailed explanation.
 - p value is an important component of the approach. There should be a proper sensitivity analysis of the p value in the experiemnts along with evaluation on complex datasets.
 - Using only tabular data to show the performance greatly reduces the applicability of the approach. Showing it for visual or text data is necessary.
 - Showing the applicability of the approach with out-of-distribution data having comparatively smaller sample size.
 - claim that using the explanation shift leading to higher sensitivity towards detecting distribution shift needs to be validated on natural datasets beyond synthetic data only.

**Strengths And Weaknesses:**

Strengths
1. The paper looks into an interesting area of analyzing model performance based on explanation shifts.
2. The authors have made considerable effort in including the different distribution shifts including their mathematical notations.
3. The authors have also attempted to theoretically analyze the sensitivity of different shifts towards model performance on new data.

Weaknesses:
1. From Eq. 1, we see that use of Shapeley values restricts the number of features to p. How does the explanation shift results vary with different p values?  Specially in complex datasets, the number of features determining the model behavior is large. The evaluation datasets have comparatively smaller feature dependencies such as UCI Adult Income dataset, which has only 14 features. This does not quite bring out the efficacy of the explanation shift. The approach should be evaluated over more complex datasets.
2. The novelty of the approach seems to be weak. It only deals with tabular data, which greatly restricts application. Also, how does the proposed approach perform over other data-types besides the tabular data. Can this approach be generalized? How does it work for non-tabular data such in case of images or language?
3. Also how does the proposed approach perform in low-data regime? The authors should evaluate the dependency of explanation shift on the scale of data samples required. Often the out-of-distribution data may have a comparatively smaller sample size, in such a case, can the approach still evaluate the distribution shift and if so to what confidence?
4. The authors claim that using the explanation shift led to higher sensitivity towards detecting distribution shift. This claim is brought forth using synthetic data only. What about in cases of natural datasets? Also, does that mean it is more prone towards error? What are the chances of false negatives and how does the sensitivity vary with different features?

---

> ### Author Response · Authors · 2024-10-26
>
> Dear reviewer,
>
> Many thanks for the comments!
>
> > From Eq. 1, we see that use of Shapeley values restricts the number of features to p. How does the explanation shift results vary with different p values? Specially in complex datasets, the number of features determining the model behavior is large. The evaluation datasets have comparatively smaller feature dependencies such as UCI Adult Income dataset, which has only 14 features. This does not quite bring out the efficacy of the explanation shift. The approach should be evaluated over more complex datasets.
>
>
> We agree with the reviewer’s observation regarding the limitation imposed by a fixed number of features, $p$. This constraint currently restricts our method’s application to tabular data, as extending it to more complex data types (e.g., vision or NLP), where the number of features varies after deployment, would require significant methodological adaptations, which fall beyond the scope of this work. This limitation is acknowledged in the main body of work. Instead, we focused on a detailed analysis within the tabular modality, as reflected in the manuscript’s depth (~30 pages), which we hope provides a strong basis.
>
> Regarding experiments with higher-dimensional data, Figure 12b includes an experiment with 25 features as a exploration of higher complexity. However, we acknowledge that calculating SHAP values becomes computationally intensive as the number of features increases.
>
> If the reviewer has specific dataset suggestions that would better reflect complex feature dependencies, we would welcome those insights.
>
>
> > Also how does the proposed approach perform in low-data regime? The authors should evaluate the dependency of explanation shift on the scale of data samples required. Often the out-of-distribution data may have a comparatively smaller sample size, in such a case, can the approach still evaluate the distribution shift and if so to what confidence?
>
> We appreciate the reviewer's point regarding low-data scenarios. In our study on Novel Group Shift, we examine cases where an increasing number of samples from an unseen group is introduced (Figures 3, 5, 6, 10, 11), which includes low-ratio regimes that may be considered "low-data" settings, where  X is out-of-distribution (OOD) and 𝑋−1 is in-distribution (ID).
>
> It is also worth noting that with very few samples, the situation may shift from OOD detection to anomaly detection. Given that our method leverages a binary classifier, techniques from imbalanced learning could be applied as an extension.
>
> If the reviewer feels this doesn’t fully address the concern, we would be glad to add further low-data experiments in the Appendix.
>
> > The authors claim that using the explanation shift led to higher sensitivity towards detecting distribution shift. This claim is brought forth using synthetic data only. What about in cases of natural datasets? Also, does that mean it is more prone towards error? What are the chances of false negatives and how does the sensitivity vary with different features?
>
> We will rephrase to "explanation shift led to higher sensitivity towards detecting distribution shift that impact the model."
> From a mathematical perspective this is demonstrated in comparison to data shifts and to prediction shift in Section 4.
>
> We agree with the reviewer that this experiment can improve the submission. False negatives can happen due to (i) issues with the calculations of the feature attributions distributions and (ii) if the model $g$ is not able to distinguish between both distributions. Aiming to solve this second issue, we propose to add in the Appendix an experiment similar to the one Figure 5, but varying the depth(complexity) of the Explanation Shift Detector.

---

> > ### Author Response · Authors · 2024-11-30
> >
> > Dear Reviewer,
> >
> > We wanted to follow up regarding our response to your feedback on our submission.
> >
> > We have aimed to address each point in our response.  We would be grateful if you could share any further thoughts, concerns, or suggestions.
> >
> > If there are specific areas of our response that you feel require additional elaboration or experimentation, we would be more than happy to address them.
> >
> > Thank you once again for your constructive feedback!

---

> > > ### Comment · Reviewer_6Tsi · 2024-12-03
> > > **Thanks for the clarifications**
> > >
> > > Thanks authors for your response to my queries. Indeed the restrictions to tabular data only and also the SHAP calculations becoming computationally intensive are really concerning. It is also worthwhile to add further low-data experiments in the Appendix. Varying the depth (complexity) of the Explanation Shift Detector would be interesting and can be included in the camera-ready.

---

### Decision · Action_Editor_hxK6 · 2024-12-15

**Recommendation:** Accept with minor revision

**Comment:**

All the reviewers recommend accepting this paper to TMLR. Here are some highlighted comments on the strengths of the paper mentioned by the reviewers:

- The concept of measuring changes in an explanation function is interesting.
- The case studies in Section 4 are useful.
- A good amount of empirical studies in Section 5 show the effectiveness.
- The authors share the work as an open-source Python package and tutorials.
- Several different distribution shifts are considered.

I also recommend acceptance with minor revision. This work makes good contributions to the methodology and its understanding to the domain of distribution shift detection, although some weaknesses are pointed out by the reviewers (see also Explain your recommendation to the ICLR Journal-to-Conference Track).

**Audience:**

The proposed framework is general and can be applied to many tasks. A large portion of researchers and practitioners working on distribution shift problems may be interested in the method and the results.

On the other hand, the authors focus on tabular data for the experiment part, which may limit the audience (also mentioned by Reviewers 6Tsi and Crub).

**Claims And Evidence:**

**Claims**:
This paper proposes a method for measuring how much impact a distribution shift has on the prediction model in machine learning tasks.
While many previous methods measure distribution shifts by comparing data distributions, the proposed method looks at the distribution of the results given by an _explanation function_ such as SHAP and LIME, which are expected to better capture the impact of the shift on predictions.

**Evidence**:
The authors provide concrete examples to highlight the properties of the proposed method.
- Example 4.1 shows a case in which comparing marginal distributions cannot detect the shift while the proposed shift measure can.
- Example 4.2 illustrates the soundness property that the explanation function does not change if the optimal prediction rule does not change. This is not the case for the comparison between the marginal distributions because it does detect the shift, although we wish to ignore this type of shift that does not affect predictions.

The authors also explain two limitations of the explanation function using examples.
- In Example 4.3, the explanation function detects the shift even though the optimal prediction rule does not change.
- In Example 4.4, on the other hand, the explanation function fails to detect the shift even though the optimal prediction rule changes.

Empirically, the experiments in Section 4.4 show that the Kolmogorov-Smirnov test performed on the distribution of the explanation function successfully rejects the hypothesis of equal distributions with convincing (very small) p-values.

Finally, Section 5 provides more experiments and empirical analyses for comparing the proposed method with other methods, and the superiority of the proposed method is observed.